# Disentangling Scatter in Long-Term Concentration-Discharge Relationships: the Role of Event Types

Felipe A. Saavedra[1], Andreas Musolff[2], Jana von Freyberg[3,4], Ralf Merz[1], Stefano Basso[1], Larisa Tarasova[1]

[1]Department Catchment Hydrology, Helmholtz Centre for Environmental Research-UFZ, Halle (Saale), 06120 , Germany
[2]Department of Hydrogeology, Helmholtz Centre for Environmental Research-UFZ, Leipzig, 04318,Germany
[3]School of Architecture, Civil and Environmental Engineering, EPFL, 1015 Lausanne, Switzerland
[4]Mountain Hydrology and Mass Movements, Swiss Federal Institute for Forest, Snow and Landscape Research (WSL), 8903 Birmensdorf, Switzerland

*Correspondence to*: Felipe A. Saavedra (felipe.saavedra@ufz.de)

**Abstract.** Relationships between nitrate concentrations and discharge rates (C-Q) at the catchment outlet can provide insights into sources, mobilization and biogeochemical transformations of nitrate within the catchment. Nitrate C-Q relationships often exhibit considerable scatter that might be related to variable hydrologic conditions during runoff events at sampling time, corresponding to variable sources and flow paths despite similar discharge rates. Although previous studies investigated the origins of this scatter in individual or in a few catchments, the role of different runoff event types across a large set of catchments is not yet fully understood.

This study combines a hydrological runoff event classification framework with low-frequency nitrate samples in 184 catchments to explore the role of different runoff events in shaping long-term C-Q relationships and their variability across contrasting catchments. In most of the catchments, snow-impacted events produce positive deviations of concentrations, indicating an increased nitrate mobilization compared to the long-term pattern. In contrast, negative deviations occur mostly for rainfall-induced events with dry antecedent conditions, indicating the occurrence of lower nitrate concentrations in river flows than their long-term pattern values during this type of events. Pronounced differences in event runoff coefficients among different event types indicate their contrasting levels of hydrologic connectivity that in turn might play a key role in controlling nitrate transport due to the activation of faster flow paths between sources and streams. Using long-term, low-frequency nitrate data we demonstrate that runoff event types shape observed scatter in long-term C-Q relationships according to their level of hydrologic connectivity. In addition, we hypothesize that the level of biogeochemical attenuation of catchments can partially explain the spatial variability of the scatter during different event types.

## 1 Introduction

Diffuse nutrient inputs in catchments are a challenge for water quality management (Paerl 1997; Stumpf et al., 2016). Excess of nutrients, such as nitrate, harms ecosystems by creating favorable conditions for eutrophication in water bodies and

leading to biodiversity loss (GEA, 2017; EEA 2019; Weitere et al., 2021). Fertilizer application on agricultural land remains the main source of nitrate contamination in human-impacted catchments, despite regulations of the past decades that stimulated a reduction of fertilizer application in Europe (Grinsven et al., 2012). Moreover, due to long-lasting legacy effects a delay in the reduction of riverine nitrate concentration was reported in many catchments (Tesoriero et al., 2013; Meter and Basu, 2017; Bieroza et al., 2018; Chang et al., 2021).

Long-term concentration-discharge (C-Q) relationships are a valuable tool for analyzing water quality gradients and trends, and for developing water management strategies (Bowes et al., 2014). The shape of C-Q relationships encodes export patterns and reflects the temporally varying quantities of critical substances such as nutrients delivered to streams (Godsey et al., 2009; Meybeck and Moatar, 2012; Rose et al., 2018). Depending on the slope of the log-log linear dependency of concentrations from discharge, three different export patterns (Godsey et al., 2009) can be defined: dilution (negative slope), enrichment (positive slope) and neutral (no relationship between C and Q or slope close to 0). Differences in long-term C-Q-relationships among catchments can be associated with differences in availability and spatial distribution of solute sources (Musolff et al., 2017; Dupas et al., 2019; Zhi et al., 2019; Casquin et al., 2021), their hydrologic connectivity (Seibert et al., 2009; Dupas et al., 2016; Covino, 2017) and biogeochemical processes within the soil and stream that can retain or permanently remove nitrate from streamwater (Mulholland et al., 2008; Dupas et al., 2016; Moatar et al., 2017; Benettin et al., 2020).

Biogeochemical processes that affect nutrient cycles in soil and water might add variability to long-term C-Q relationships. The effectiveness of the denitrification process, which removes nitrate from the soil, depends on periodic environmental factors such as temperature and soil moisture and the availability of electron donors (Korom et al., 2012; Ortmeyer et al., 2021). Instream removal processes are also more efficient during low flows and higher temperatures, adding more variability to the low-flow portion of the long-term C-Q relationships (Dehaspe et al., 2021; Moatar et al., 2017). Moreover, the availability of nitrate sources is balanced by fertilizer application and mineralization of organic nitrogen compounds and hence varies in time adding temporal variability to C-Q relationships. Timing of fertilizer application is often unknown, and the mineralization processes depend on chemical soil conditions and environmental factors (e.g., soil moisture and temperature) that mediate communities of microorganisms (Curtin et al., 2012; Guntiñas et al., 2012). Average residence times of nitrate in agricultural catchments can last for decades, producing a legacy in soil (Meter et al., 2016; Puckett et al., 2011; Tesoriero et al., 2013; Vervloet et al., 2018) that can buffer the periodic effect of biogeochemical processes reducing the variability in the concentration of nitrate (Basu et al., 2011; Bieroza et al., 2018; Thompson et al., 2011).

The scatter of C-Q relationships might also be related to hydrologic conditions at the time of sampling (Knapp et al., 2020, Musolff et al., 2021), which are investigated for a large number of catchments only by a few recent studies (Minaudo et al., 2019; Pohle et al., 2021). Minaudo et al. (2019) showed that in most of the 219 French catchments nitrate samples taken

during baseflow conditions exhibit an enrichment export pattern, while during runoff events a neutral or opposite pattern (dilution) prevail, generating scatter in the combined long-term C-Q relationships. The cause of this scatter can be also traced to a variety of responses observed at the event-scale in several studies with high-frequency data in single or a few catchments (e.g., Bowes et al., 2015; Lloyd et al., 2016; Koenig et al., 2017; Gorski and Zimmer, 2021).

Our study relies on low-frequency nitrate data, which is often used to build long-term C-Q relationships (e.g. Cartwright et al., 2020, Diamond and Cohen 2018). However, studies with high-frequency data found large variability in the C-Q patterns during events (e.g. Knapp et al., 2020; Dupas et al., 2016; Vaughan et al., 2017) that might add scatter to the long-term C-Q relationship. Disparate patterns of the event C-Q relationships in a catchment over time are mainly attributed to varying dominant flow sources (e.g., groundwater, shallow subsurface flow), antecedent wetness conditions (Inamdar et al., 2006;
Knapp et al., 2020; Vaughan et al., 2017), time of fertilizer application (Bowes et al., 2015; Dupas et al., 2016; Outram et al., 2016), biogeochemical cycling (Heathwaite and Bieroza, 2021) and runoff event characteristics or types (Butturini et al., 2006; Bauwe et al., 2015; Chen et al., 2020; Knapp et al., 2020). For example, Winter et al. (2022) showed in a few catchments located in Central Germany that runoff events generated by rainfall with dry antecedent conditions export lower nitrate concentrations due to lower hydrologic connectivity but exhibit a high variability of event C-Q slopes. In contrast,
Knapp et al. (2020) showed using high-frequency concentration and discharge observations from one small forested catchment located in Switzerland that during larger runoff events with dry antecedent conditions the slopes of the event-scale. C-Q relationships are more positive due to the accumulation of nitrate in the soil during dry periods by atmospheric deposition and the subsequent mobilization by event water. Moreover, in several catchments in US and Europe, snow-induced events were found to export high nitrate concentration (Koenig et al., 2017; Inamdar et al., 2006; Casson et al.,
2014). Similarly, in the previously mentioned Central German catchments Winter et al. (2022) found high nitrate concentrations and flat event C-Q slopes during snow-impacted events indicating that sufficient nitrate sources are available and most of the relevant flow paths are activated and connected to the stream during such events.

It was shown that hydrologic connectivity as a portion of the catchment connected to the stream via surface or subsurface
pathways, increases according to the wetness state of the catchment (Blume & Van Meerveld, 2015, Jencso et al., 2009) and modulates export of nutrients at different scales. At seasonal scale nutrient transport to streams can be increased with higher hydrologic connectivity in catchments with abundant sources (Martin et al., 2004; Veith et al., 2020; Guillemot et al., 2021).. At event scale the activation of different flow paths during different levels of hydrologic connectivity evaluated using shallow wells or models can partially explain changes in nitrate concentration during events (von Freyberg et al., 2014,
Ocampo et al., 2006; Stieglitz et al., 2003). However, at the larger scale such observations are not available.

At catchment scale soil moisture or discharge rates are often used as proxy of hydrologic connectivity (e.g., Bracken et al., 2013; Jencso et al., 2009). Event runoff coefficient (i.e., a volumetric ratio of quick flow and input precipitation or

snowmelt), which represents how efficiently streamflow responds to catchment water inputs, can be also considered as its proxy (e.g., Blume et al., 2007; Outram et al., 2016; von Freyberg et al., 2014). Higher runoff coefficients are associated with wetter antecedent catchment states, indicating that such conditions favor a more efficient rainfall-runoff response (Tarasova et al., 2018, Outram et al., 2016) and possibly activation of more surface and subsurface hydrologic flow pathways that facilitate fast transport of water and nutrients from the landscape to the stream (Blume & Meerveld, 2015; Hardie et al., 2011; Stieglitz et al., 2003).

New approaches to characterize and classify runoff events according to hydrologic conditions offer a possibility to efficiently aggregate information about antecedent wetness state of catchments and characteristics of inducing events (e.g., rainfall, snowmelt) and to distinguish events with contrasting hydrological responses for large number of catchments (Tarasova et al., 2020). Such classification of event types combined with concentration of nitrate in stream water might unravel scatter in long-term C-Q relationships as exemplified in Fig. 1. In Fig. 1a biweekly nitrate data are associated with the event type at the time of stream water sample collection. When these data are plotted in the log-log C-Q space (Fig. 1b) some event types exhibit positive (higher concentration) or negative (lower concentration) deviations from the long-term C-Q relationship. Our study aims for the first time to investigate the presence of systematic deviations in long-term C-Q relationships produced by different runoff event types for a large dataset of catchments.

We hypothesize that these deviations are related to the differences in nitrate transport during these event types and we investigate such deviations from the long-term C-Q relationships in 184 German catchments. Specifically, our goal is to examine the effect of runoff event types on the observed scatter in C-Q relationships by addressing the following research questions:

1 Do samples collected during different event types deviate differently from the long-term C-Q relationships observed at the catchment outlets?

2 Which climatic and landscape characteristics explain differences in the observed C-Q deviations among German catchments?

3 Which are the potential mechanisms that explain the direction and magnitude of C-Q deviations for different event types?

Understanding the nature of nitrate deviations from the long-term C-Q relationship might provide useful information for water quality managers to reduce the risk of extreme nitrate loads to water bodies, as well as improve sampling campaigns to better capture nitrate C-Q scatter.

## 2 Methods

### 2.1 Study catchments and data

In this study we analyzed low frequency (biweekly to monthly) nitrate concentration data from 184 mesoscale catchments in Germany for the period from 2000 to 2015. The data were obtained from the water quality and quantity database of Germany (Musolff, 2020; Ebeling et al., 2021) in combination with a recently developed classification framework of runoff events (Tarasova et al., 2020). Similar to Ebeling et al. (2021) we exclude the data prior to the 2000s to avoid impacts of improved wastewater treatment technologies in Germany. In total, we consider 33,713 nitrate samples.

Sizes of study catchment range from 95 to 23,615 km2 (with a median size of 704 km²) and cover all four main German natural regions: the North German Plain, Central Uplands, South German Scarplands and Alpine Forelands (Fig. 2a). The climate varies from temperate oceanic to temperate continental from West to East. Mean annual precipitation ranges from 567 mm in the Lowland northeastern catchments up to 1379 mm in the alpine catchments in the South. The predominant land use in the study catchments is agriculture, with a median coverage among catchments of 50% and a range from 13% to 84%. Median portion of catchment area covered by forest is 41% of catchment area (Fig. 2b).

The runoff event classification framework of Tarasova et al. (2020) considers runoff events identified from daily discharge data in catchments with no major flow regulations. The location of the discharge stations does not always coincide with water quality stations in the dataset of Ebeling et al. (2021). Both data sets are linked by pairing stations that are located on the same stream and differ less than 20% in their drainage areas. These were considered as identical outlets similarly to Guillemot et al. (2021). The mean overlap between drainage areas of the corresponding outlets from the two datasets is 95% with a standard deviation of 5%.

### 2.2 Identification and classification of hydrological events

Runoff events and corresponding precipitation events were separated using an automated time series approach developed by Tarasova et al. (2018). The method was applied to daily discharge and precipitation data obtained from the REGNIE data set (Rauthe et al., 2013). The method includes baseflow separation, precipitation attribution (i.e., corresponding inducing rainfall and/or snowmelt events are linked to runoff events) and an iterative procedure to adjust site-specific thresholds for the refinement of multi-peak events. The median event duration is 12 days with a standard deviation of 7.7 days. The shortest event duration is one day, however 95% of the identified events exhibit a duration of 3 or more days. Each identified runoff event was then classified considering in the first place the nature of inducing events (rainfall, mixture of rainfall and snowmelt or rain-on-snow) (Fig. 3a) using the proportions of rainfall and snowmelt in the total volume of precipitation events (Table S1). In the second step, we considered the antecedent wetness state (wet or dry) by accounting for catchment-

averaged soil moisture state prior to the event. Catchment-average snow water equivalent and soil moisture were simulated by the mHM model (Samaniego et al., 2010; Kumar et al., 2013) and provided in Zink et al. (2017). Additionally, the classification considers spatial organization of soil moisture within the catchment using spatial coefficient of variation of soil moisture, classifying events as uniform or patchy, with the latter corresponding to highly variable soil moisture within the catchment. A more detailed description of the classification framework is provided in Tarasova et al. (2020).

Each nitrate sample was linked to either no event (No.event), or to one of the five event types (Fig. 3a): rain-on-snow (Rain.on.snow), mixture of rainfall and snowmelt (Mix), rainfall during wet antecedent conditions (Rain.wet), rainfall during dry antecedent conditions with spatial uniform distribution of soil moisture (Rain.dry.uniform) and rainfall with dry antecedent condition with heterogeneous spatial distribution of soil moisture (Rain.dry.patchy). Note that we simplified the event types to increase the number of nitrate samples of each event type.

## 2.3 Long-term C-Q export patterns

For each catchment, the long-term C-Q relationship was derived as a linear regression between nitrate concentration (C) and discharge (Q) in the log-log space (Fig. 3b). Based on the slope of the long-term C-Q relationships (b), we grouped all study catchments according to three different long-term C-Q export patterns: dilution (b<0.1) refers to a limitation of sources during high flows, enrichment (b>0.1) is related to a transport limitation with abundant sources or solute uptake during low flows (Moatar et al., 2017) and neutral (b~0) indicates no monotonic relationship between C and Q. As stated by Ebeling et al. (2021) this latter group exhibits largely invariable concentration with low ratios of coefficients of variation (CVc/CVq). Three different catchments are shown as an example of each export pattern in Fig. 3c.

## 2.4 Quantifying the deviations from long-term C-Q relationship

For each catchment we want to quantify if samples taken at a specific event type show systematic deviations from the long-term C-Q regression compared to all samples. We quantified the deviation of each grab sample from the long-term C-Q relationship for each catchment by computing the corresponding residual concentration from the long-term C-Q linear regression line (Fig. 3b). Resulting residuals were subsequently grouped according to the hydrological event type at the time of sampling.

Due to the variable number of grab samples attributed to different event types (Fig. S1), for each catchment, we performed a bootstrapping procedure that can explicitly handle unbalanced data by iteratively comparing two random subgroups of samples with the same size (undersampling method, e.g., Branco et al., 2015). The procedure is implemented in the following way for each catchment: $n$ nitrate samples of a certain event type and the same number of nitrate samples from all samples (general pattern) are chosen randomly with replacement (i.e., each data point can be chosen more than once, following bootstrapping procedure). The difference of median residuals of an event type and residuals of the general pattern

is then the measure of deviation of a corresponding event type from the long-term C-Q relationship (Δres). We obtained this measure 10,000 times to robustly compute its distribution (Fig. S2) and median value (Δres50). The number of samples $n$ was chosen for each catchment and event type according to the number of nitrate samples available for the corresponding event type. For each catchment, event types with less than 10 nitrate samples are excluded from the analysis. The median number of nitrate samples among all study catchments and event types is 27.

For each catchment and from all the iterations, we obtain the median deviations between event types and the general pattern (Δres50). In order to evaluate the persistence of C-Q deviations across catchments, we tested the significance of Δres50 across catchments for each event type using the non-parametric Kruskal-Wallis test (Kruskal & Wallis, 1952) at the significance level α=0.05.

Low frequency datasets such as the one used in our study might contain samples collected during different phases of the event hydrograph (e.g., falling or rising limb). This might hamper the interpretability of the results due to possible bias in observed nitrate concentration linked to the time of sampling and the hysteresis effect revealed in high-frequency observations (e.g., Lloyd et al., 2016; Vaughan et al., 2017). In fact, Pohle et al. (2021) showed systematic differences in nitrate concentration between samples collected during rising and falling limbs for numerous catchments in Scotland. To understand the potential effect of the hysteresis on the deviations from long-term C-Q (Δres50) we repeat the bootstrapping procedure described above considering samples collected during the rising limb, falling limb and near the event peak (near-to-peak). The rising limb of a runoff event starts at the beginning of the event and finishes one day before the day of the peak discharge. The falling limb starts one day after the day of the peak discharge and finishes at the end of the runoff event. In addition, we defined near-to-peak as samples collected from one day before to one day after the day of the peak discharge. Of the total samples taken during runoff event types 34% correspond to the rising limb, 55% to the falling limb and 30% to near-to-peak. Notice that definition of near-to-peak samples allows some overlap with the other two groups of samples to use a more balanced number of samples than considering samples collected on the day of the peak of discharge only (11% of the samples were collected during the day of the peak discharge).

**2.5 Catchment descriptors and relationships to C-Q deviations**

In order to explore the differences of deviations from the long-term C-Q relationship across the catchments, we examined the Spearman rank correlation of median residuals for each catchment with various catchment descriptors. Here, we only examine catchment descriptors that were previously identified as primary controls of the nitrate C-Q export patterns in Germany (Ebeling et al., 2021). This includes topographic descriptors: median topographic wetness index, median slope and area; land cover descriptors: fraction of agriculture, forest and artificial surface; soil and aquifer descriptors: median soil depth and fraction of sedimentary aquifer; nitrate sources descriptors: nitrate surplus, agricultural horizontal heterogeneity, nitrate vertical ratio; and hydrometeorological descriptors: aridity index, mean annual potential evapotranspiration,

precipitation and temperature (Table S2). Detailed derivation of the above-mentioned catchment descriptors is provided in Ebeling et al. (2021).

## 3. Results

### 3.1 Frequency of runoff event types

Stream water samples taken during runoff event conditions account for 58% of all samples. These samples are classified to
230 one of the five event types: 18% to Rain.dry.patchy type, 11% to Rain.dry.uniform, 15% to Rain.wet, 7% to Rain.on.snow and 7% to Mix.

On average across catchments, the fraction of samples taken during each event type vary at different discharge rates. Above median discharge rate, 74% of all samples correspond to an event and event types Rain.wet, Rain.on.snow and Mix occur
more frequently (Fig. 4a). In contrast, only 49% of samples below median discharge rate were taken during an event and most of these grab samples correspond to Rain.dry.patchy and Rain.dry.uniform types.

The frequency of event types also varies seasonally (Fig. 4b). In winter most of the grab samples were taken during Rain.on.snow, Mix and Rain.wet event types. In spring months Rain.dry events become more frequent than Rain.on.snow,
Mix and Rain.wet event types. During summer, most of the samples were taken either under No.event conditions or during Rain.dry.uniform and Rain.dry.patchy events. In autumn, the frequency of grab samples taken during Rain.wet, Rain.on.snow and Mix event types increases.

### 3.2 Long-term C-Q relationships and deviations during event types

We computed long-term nitrate C-Q relationships for the 184 catchments, obtaining slopes (b) from -0.6 to 1.48, with a
245 mean of 0.13. In total, 88 study catchments exhibit neutral patterns, 80 catchments are characterized by enrichment patterns and only 16 catchments show dilution patterns. Across all catchments, the median $R^2$ value of the long-term C-Q relationships was low (0.14), indicating presence of considerable scatter in the regressions.

We explored the residuals (res) of all nitrate data from all catchments, finding that 65% and 68% of the samples taken during
Rain.on.snow and Mix event types have positive residual values respectively, indicating that concentrations were higher than the long-term log-log linear C-Q regression. In contrast, during Rain.dry.patchy and Rain.dry.uniform 69% and 60% of the samples have negative residuals. We found a less clear picture for samples taken during Rain.wet events and No.event conditions with 53% and 56% of residuals positive respectively (Fig. S3).

We found strong differences in median deviations from the long-term C-Q relationships (Δres50) among different event types (Fig. 5a). Rain.on.snow and Mix event types have more often positive Δres50 values (79% and 93% of the study catchments correspondingly) (Fig. 5b) when comparing across catchments. Instead, Rain.dry.patchy and Rain.dry.uniform event types show negative values of Δres50 more often (96% and 61% of the study catchments), with Rain.dry.patchy events showing stronger deviations. Contrasting behavior between snow-impacted events (i.e., Mix and Rain.on.snow) and rainfall events with dry antecedent wetness conditions (Rain.dry.patchy and Rain.dry.uniform) occurs across most of the study catchments independently from their long-term export pattern (Fig. 5b). For Rain.wet events deviations can be negative as well as positive (52% and 48% of study catchments respectively) with a median of Δres50 across catchments close to zero (Fig. 5a). For samples that were taken during No.event conditions, Δres50 value is slightly positive in 85% of all catchments.

The sign of C-Q deviations are in line with observed nitrate concentration during different event types (Fig. S4). Negative residuals during Rain.dry.patchy and Rain.dry.uniform coincide with lower nitrate concentrations for most of the catchments independent of the long-term C-Q pattern. Similarly, during Rain.on.snow and Mix events positive C-Q deviations correspond to nitrate concentrations higher than median for most of the catchments with neutral or enrichment C-Q pattern. For catchment with dilution export pattern, nitrate concentration for Rain.on.snow and Mix events is similar to the average, however higher discharge in this case generate positive residuals.

We analyzed the influence of the sampling time within runoff events separating samples taken during the rising limb, near to the peak and falling limb. Although there are certain data limitations for a few groups of samples (gray tiles in Fig. S5b), we are able to reproduce the analysis for most of the cases. Similarly to the case when using all samples (Fig. 5b), the values of Δres50 for samples taken during the rising limb, near to the peak and falling limb are mostly positive for Rain.on.snow and Mix events, negative for Rain.dry.patchy and Rain.dry.uniform. Our results confirm that the time of sampling during runoff events does not affect our findings regarding median C-Q deviations for different types of runoff events.

Although the sign of C-Q deviation is consistent across catchments for most of the event types, the magnitude of deviation varies across catchments (Fig. 5a). The variability of Δres50 expressed as interquantile ranges across catchments (boxplots in Fig. 5a) is the lowest for the samples taken during no event conditions (0.03) and Rain.wet events (0.06). The largest variability was detected for Rain.dry.patchy events (0.1), followed by Mix (0.09) and Rain.dry.uniform events (0.09).

### 3.3 Variability of C-Q deviations across German catchments

We analyzed the spatial variability of C-Q deviations for different event types (Fig. 5a) computing Spearman rank correlations between deviations and catchment descriptors. We found significant correlations between Δres50 for each event type and catchment descriptors. Topographic properties (i.e., median slope and topographic wetness index) have the strongest correlation to the Δres50 values of almost all event types (Fig. 6). Specifically, flatter catchments (low median

topographic slope) with greater soil depths that are mostly located in the Northern Germany and Alpine Foreland tend to exhibit more positive residuals for Rain.wet, Rain.on.snow and Mix events, and more negative residuals for Rain.dry.patchy events and samples taken during no event conditions (Fig. 5a). Catchments with these characteristics often show high agricultural land cover (Fig. S6), however the fraction of agriculture show less significant correlations with Δres50 than topographic descriptors. Moreover, in catchments with larger fractions of water-impacted soils (e.g., stagnosols, semi-terrestrial, semi-subhydric, subhydric and moor soils) we found more positive residuals for snow-impacted events (Rain.on.snow, Mix) and more negative residuals for Rain.dry.patchy events. These catchments are often located in Central East or North West Germany.

Correlations between Δres50 and fraction of agriculture in the catchments are less significant than those with topographic descriptors (Fig. 6). Instead, we observed strong correlations between Δres50 and the fraction of forest (p<0.01). Forested catchments show less positive Δres50 values for Rain.on.snow events and less negative values for Rain.dry.patchy. However, we also noticed that the fraction of forest is positively correlated with topographic slope and negatively correlated with soil depth and the fraction of agriculture (Fig. S6).

Nutrient source descriptors were also significantly correlated with Δres50. Horizontal heterogeneity of agricultural sources correlates negatively with Rain.dry.pachy residuals and vertical concentration ratio of nitrate correlates negatively with Δres50 values of Rain.dry.patchy and No.event conditions. Nitrate surplus is significantly related only to Rain.wet residuals.

### 3.4 Relationship between hydrologic connectivity and event type variations in residuals

We examined event runoff coefficients corresponding to different catchments and event types to link the relation between hydrologic connectivity for these event types and corresponding deviations of their samples from the long-term C-Q relationship (Fig. 7a). Catchment median event runoff coefficients exhibit a coefficient of variation of 41% across catchments. Nevertheless, variability of median runoff coefficients across event types for single catchments is larger in most of the cases, with coefficients of variation from 12% to 118% and a median value of 67% across catchments. We found that event types with significantly higher median runoff coefficients also exhibit significant differences in Δres50 values (Fig. 7b and 7c). Only Mix and Rain.on.snow events have similar runoff coefficients and similar Δres50 values.

## 4. Discussion

### 4.1 Direction and magnitude of C-Q deviations for different event types

We found systematic differences in the direction and magnitude of deviations of nitrate concentrations (Δres50) from the long-term C-Q relationship during different types of runoff events despite the large variety of study catchments (Fig. 5). In

the following paragraphs, we will discuss potential mechanisms that can explain the variability of C-Q deviations across event types.

320

Positive deviations for nitrate concentrations during snow-impacted events (i.e., higher nitrate concentration compared to the general C-Q pattern) are in line with previous studies that have shown an increase of nitrate concentration in stream water during snow-impacted events in forested and agricultural catchments (Inamdar et al., 2006; Casson et al., 2014; Koenig et al., 2017). This is in line with Winter et al. (2022), who showed using high-frequency data that snow-induced events export the highest nitrate concentration compared to other event types in six German catchments with mixed land use. Our results clearly show that snowmelt does not generate lower concentration of nitrate compared to the long-term C-Q relationship, although this might be expected due to lower nitrate concentration in snowfall than in stream water from agricultural catchments (Johannsen et al., 2008). Instead, higher concentration indicates enhanced nitrate transport from soil sources with no source limitation during these types of events. We argue that during snow-impacted events hydrologic connectivity between nitrate sources and streams is high due to elevated wetness conditions (Stieglitz et al., 2003) that is consistent with previously reported high nitrate concentration during the winter period (Martin et al., 2004; Ocampo et al., 2006; Yang et al., 2018). Due to excessive catchment wetness during snow-impacted events, a high amount of new water transported by faster and shallower pathways can reach the stream (a so-called inverse storage effect; Fang et al., 2019) mobilizing large amounts of nitrate available in the soil (Yang et al., 2018). In adition, the mobilized water during these events is less affected by biogeochemical processes due to lower microbial activity induced by low temperature during snow-impacted events (Johannsen et al., 2008).

Furthermore, our analysis shows that Rain.dry.uniform and Rain.dry.patchy events generate lower nitrate concentrations compared to the other types of events or no event conditions (Fig. S4), producing strong negative C-Q deviations (Fig. 5). Along the same lines, Winter et al. (2022) showed that runoff events with dry antecedent conditions exhibit lower concentration compared to other event types in six German catchments with mixed land use. There are two possible explanations for the occurrence of this phenomenon. On one hand, Rain.dry.uniform and Rain.dry.patchy events occur more often during the dry season, when nitrate concentrations are reported to be lower (House et al., 2001; Guillemot et al., 2021) due to a hydrological disconnection between agricultural sources and streams under dry conditions, as well as higher biogeochemical nitrate removal processes, including biotic uptake and denitrification (Mulholland et al., 2008; Rode et al., 2016, Lutz et al., 2020, Johannsen et al., 2008). On the other hand, during runoff events with dry antecedent conditions nitrate concentrations can be diluted below pre-event concentration levels. This is shown by high-frequency observations in agricultural catchments that report more frequent negative event C-Q slopes during the dry season (Winter et al., 2021, 2022; Zhang et al., 2020; Koenig et al., 2017). In such cases nitrate concentration decreases compared to pre-event concentrations due to hydrologic disconnection between streams and agricultural land, and growing importance of runoff generated from riparian zones (Fang et al., 2019; Grayson et al., 1997; McGlynn & Seibert, 2003), which are known to buffer nitrate inputs

due to high denitrification potential (Ocampo et al., 2006; Cole et al., 2020; Sabater et al., 2003). Our results show that the combined effect of lower pre-event concentration and further decrease in concentrations due to runoff events magnifies the observed negative deviations of nitrate samples from the long-term C-Q relationships. Nevertheless, the data available for this study does not allow to quantify the contribution of individual effects of these two factors on the scatter of long-term C-Q relationships of nitrate. On the contrary, studies in pristine headwaters and forested catchments found that rainfall events with dry antecedent conditions can mobilize large amounts of nitrate increasing the concentration in streams (Knapp et al., 2020; Koenig et al., 2017). Since these findings are based on the observations in a single or only a few catchments with limited agricultural activity, different nitrate sources, such as atmospheric deposition or nitrate fixation and nitrate accumulation in soil between events, might be more relevant. Agriculture is a dominant land use type in the catchments used in this study (median fraction of agricultural land is 50%), therefore a considerable nitrate accumulation in soil as the result of fertilization dominates over any other nitrate source (Häussermann et al., 2020; Lassaletta et al., 2014), explaining the discrepancy between our findings and the results from pristine headwaters and forested catchments on the role of rainfall events with dry antecedent conditions for nitrate mobilization.

Differently from runoff events with dry antecedent conditions, we found that nitrate grab samples taken during no event conditions exhibit slightly positive deviations, indicating higher concentrations compared to the long-term C-Q relationships. No event samples also exhibit higher nitrate concentrations (Fig. S4) compared to Rainfall events with dry antecedent conditions (i.e., Rain.dry.patchy and Rain.dry.uniform), with both groups of samples being collected during relatively low discharge conditions (Fig. 4a). This suggest that the lack of dilution during no events might produce more positive residuals.

## 4.2 The role of hydrologic connectivity between different event types

The hypothesized role of hydrologic connectivity on shaping nitrate deviations during runoff events is supported by the relation between event runoff coefficient and the deviation of nitrate concentrations from the long-term C-Q relationship for different event types (Fig. 7). Higher runoff coefficients indicate a more efficient rainfall-runoff response, either due to the activation of stored water or the fast runoff of rainwater or snowmelt into the stream. Across all studied catchments the highest runoff coefficients are consistently found for snow-impacted events (Rain.on.snow and Mix) (Fig. 7). High values of runoff coefficients were connected to highly positive residuals indicating that compared to the C-Q relationship more nitrate was mobilized during high levels of hydrologic connectivity. Studies using high-frequency data show that during runoff events in wet seasons, when catchments are hydrologically more connected, shallow flow paths are activated transporting greater amounts of nitrate (Inamdar et al., 2006; Outram et al., 2016; Schwientek et al., 2013). Similarly, von Freyberg et al. (2014), Ocampo et al. (2006) and Stieglitz et al. (2003) showed that upland zones are more efficiently connected to riparian zones by shallow pathways during wet months, permitting effective transport of nutrients to the stream.

There are no significant differences in event runoff coefficient between two types of snow-impacted events (i.e., Mix and Rain.on.snow) are (Fig. 7), despite possible differences in their characteristic snowmelt intensities (Tarasova et al., 2020). While during Mix events the melting of the snowpack is induced by temperature increase only, during rain-on-snow events additional portions of snowpack might be melted by additional energy brought by rainfall (Cohen et al., 2015). However, in both cases event runoff coefficients are similarly high (Fig. 7b) suggesting similar hydrologic connectivity (Blume & Meerveld, 2015). This, in turn, results in similar deviations of nitrate concentrations from the long-term C-Q relationships for these two snow-impacted event types, again highlighting the primary role of hydrologic connectivity on the observed deviations of C-Q relationships.

In contrast to snow-impacted events, lower runoff coefficients typical for rainfall events with dry antecedent conditions (i.e., Rain.dry.patchy and Rain.dry.uniform) indicate that a small portion of event water can reach the stream, which means that distant zones from the stream network associated with longer pathways exhibit lower or lack of connection during these types of runoff events. Moreover, the overall dry antecedent conditions with a heterogeneous spatial distribution of soil moisture indicate a potential disconnection of runoff generation zones, and therefore the hydrologic connectivity might be lower during such events (i.e., Rain.dry.patchy) than during events with uniform spatial distribution of soil moisture (i.e., Rain.dry.uniform) as shown by lower event runoff coefficients of the former (Fig. 7). Differences in the connectivity of these two types of events are also in line with differences in residuals with the strongest negative residuals observed for Rain.dry.patchy events. Similarly, Outram et al. (2016) showed that lower event runoff coefficients during runoff events with dry antecedent conditions mobilize only a small quantity of nitrate due to the inactive subsurface pathways. Studies using high-frequency nitrate and discharge data showed that during dry periods upland and riparian zones are usually disconnected (Ocampo et al., 2006; Stieglitz et al., 2003; von Freyberg et al., 2014). These evidence from single catchments are in line with our results across a large set of German catchments suggesting a crucial role of hydrologic connectivity for nutrient transport.

During Rain.wet events runoff coefficients are between those of snow-impacted events and rainfall events with dry antecedent conditions (Fig. 7) which indicate intermediate level of hydrologic connectivity between sources and streams, and thus both positive and negative C-Q deviations (Fig. 5).

### 4.3 Climatic and landscape controls of the variability of C-Q deviations across Germany

Despite systematic differences of C-Q deviations for different event types, we found considerable spatial variability in the magnitude of these deviations across German catchments (Fig. 5a). In the next paragraphs, we discuss how catchment characteristics may control deviations of concentrations taken during events from the long-term C-Q relationship (Δres50).

The correlation of C-Q deviations during snow-impacted events with topographic and soil properties (Fig. 6) indicates that in flatter catchments with thick soils and a high fraction of sedimentary aquifers these types of events generate higher nitrate concentrations compared to the average long-term behavior. Contrarily, C-Q deviations during Rain.dry.patchy are more negative indicating that nitrate concentrations during these events tend to be smaller than the long-term average in catchments with these characteristics. Previous studies have shown how these characteristics are able to promote nitrate removal in catchments. Deep sedimentary aquifers have a high potential of denitrification due to a great availability of electron donors, longer transit times and more anoxic conditions due to sufficient reduction capacity (Kunkel et al., 2004; Wendland et al., 2008; Knoll et al., 2020) generating a lower nitrate supply in deeper soils compared to shallow soil (Dupas et al., 2016). In addition, flat catchments (low topographic slope, higher topographic wetness index) tend to have a higher portion of riparian wetlands (Musolff et al., 2018) that can reduce nitrate concentrations from stream water. During snow-impacted events fast flow pathways between nitrate sources and the stream are activated and nitrate stored in shallow soils can be mobilized bypassing the denitrification attenuation of the soil and the riparian zone, which is also largely suppressed during low temperatures in winter (Johannsen et al., 2008; Lutz et al., 2020), resulting in higher nitrate C-Q deviations. In contrast, Rain.dry.patchy events might mobilize water from connected near-stream source zones, where riparian wetlands from flatter areas contribute water with low nitrate concentration (Fovet et al., 2018; Lutz et al., 2020) generating more negative C-Q deviations. In addition, deviations during these events might be associated with longer transit times due to thicker soil and less hydrologic connectivity (Yang et al., 2018) which can reduce nitrate concentrations in streams. Instead, steeper catchments with shallow soils during Rain.dry.patchy events show less nitrate attenuation due to shorter flow paths and less favorable conditions for denitrification, generating relatively higher streamwater nitrate concentrations during these events and therefore decreasing the magnitude of C-Q deviations.

We acknowledge that some catchment characteristics are highly correlated (Fig. S6). Flatter catchments often exhibit higher fractions of agriculture, therefore more diffuse source availability. Although the correlation of the fraction of agriculture and C-Q deviations during Rain.on.snow events was less significant than topographic descriptors, a potential increment of diffuse sources in flatter catchments might also enhance the mechanism of nitrate bypassing the buffer capacity of catchments during Rain.on.snow events generating higher C-Q deviations. For most of the event types we found that the fraction of agriculture itself is not sufficient to explain the differences in nitrate deviations from the long-term C-Q relationship between catchments (Fig. 6). However, the vertical ratio of nitrate between topsoil and groundwater and horizontal spatial distribution of agricultural land within the catchment (i.e., horizontal heterogeneity) were strongly correlated with C-Q deviations for Rain.dry.patchy events (Fig. 6). During Rain.dry.patchy events the mobilization of distant nitrate sources (horizontally and vertically) is reduced due to the low hydrologic connectivity resulting in lower nitrate concentration of stream water and more negative deviations in catchments with top loaded nitrate profile as well as with more distant agricultural lands from streams. The high spatial variability of agricultural nitrate sources, expressed as horizontal heterogeneity and vertical ratio of nitrate, and the temporal variability of sources possibly induced by elevated

450    subsurface and riparian zone removal during different levels of hydrologic connectivity, promotes deviations of nitrate concentrations from the long-term C-Q relationships.

## 4.4 Implications of this study

In this study we performed the first large-scale analysis of long-term nitrate C-Q relationships differentiating runoff event types. We show that flatter catchments with soil conditions favorable for denitrification or distant nitrate sources are prone to 455    generate disproportional loads during runoff events with high levels of hydrologic connectivity, presenting an ecological risk for aquatic ecosystems. These findings can be instructive for implementing more effective water quality management strategies to prevent extreme nitrate loads reaching water bodies in such catchments during events associated with high levels of hydrologic connectivity (i.e., snow-impacted events).

The connection between nitrate concentrations and different types of runoff events shown in our study indicates that possible changes in the occurrences of different event types due to the ongoing climate change might in turn affect the dynamics of nutrient exports in the catchments. With advancing climate change air temperature is projected to increase further leading to a substantial decline in seasonal snowpack accumulation and earlier snowmelt onset in Central Europe (IPCC, 2021). Several studies reported a reduction in snow accumulation in Germany over the last decades (Fontrodona Bach et al., 2018; Chan et 465    al., 2020; Taszarek et al., 2020) with a consistent reduction in the frequency of rain-on-snow events (Cohen et al., 2015), suggesting that the corresponding positive deviations from the long-term nitrate C-Q relationship are likely to occur less often in the future. Less frequent snow-impacted events would reduce nitrate mobilization from the soil under these critical event conditions. Consequently, more nitrate may remain in the soil sources. A fraction of this soil nitrate is expected to be removed by denitrification whereas another fraction may last longer as soil nitrate legacy (Dupas et al., 2020; Meter et al., 470    2016), thus generating unknown long-term effects in the nitrate dynamics during future runoff events. On the other hand, higher temperatures lead to a decrease of soil moisture (Dai et al., 2004), propitiating dry conditions and reducing hydrologic connectivity. An increase in frequency of rainfall events with dry antecedent conditions observed in several German catchments (Winter et al., 2022) indicates that negative deviations might become even more frequent during warm seasons in the future.

By using low-frequency, long-term nitrate data we were able to provide information about characteristic nitrate transport during different types of events and to identify hydrologic connectivity associated with these types as a critical control of nitrate dynamics in German catchments. Our findings using low-frequency data are largely supported by the detailed analysis of high-frequency data in individual catchments from the previous studies: but thanks to the large number of 480    analyzed catchments allow for a more comprehensive analysis of systematic deviations of nitrate concentrations during events of different types and provide valuable insights on the origins of the scatter in C-Q relationships. The abundance of low frequency data worldwide and transferable nature of the applied event classification framework provide the means of

further applications in contrasting environments to better understand nitrate long-term C-Q relationships across contrasting environments. Moreover, our results suggest that sampling campaigns should be designed specifically to capture runoff

events with different levels of hydrologic connectivity in order to better explain the scatter in long-term C-Q relationships and better isolate the role of singular processes (i.e., nitrate uptake, denitrification).

Although the presence of the event-scale hysteresis effect might considerably affect nitrate concentration during rising and falling limbs of the event hydrograph in some catchments (Pohle et al, 2021) we found a similar direction of deviations from

490 the long-term C-Q relationships when we considered samples taken during rising limb, falling limb and near to the peak (Fig S6b). Hence, our results suggest that the variability potentially added by the presence of hysteresis patterns is lower than the deviations observed for different event types from the long-term C-Q relationship. Increasing availability of high-frequency datasets coupled with new statistical modeling approaches might be used in the future to evaluate hysteresis-related effects in the existing long-term C-Q datasets to further disentangle inter- and intra-event variability of nitrate dynamics at larger

scales.

## 5. Conclusions

We analyzed for the first time the effect of different runoff event types on the scatter observed in concentration-discharge (C-Q) relationship across 184 German catchments. Specifically, we examined the deviations of the concentration of nitrate samples collected during different runoff event types from the long-term C-Q relationships. Our results highlight pronounced

deviations in most of the catchments regardless of their overall long-term C-Q export patterns (dilution, neutral, or enrichment). Thus, scatter apparent in long-term C-Q relationships can indeed be partially explained by different types of runoff event conditions.

We found that nitrate transport is enhanced during snow-impacted events compared to long-term C-Q relationships. On the

505 other hand, nitrate concentrations tend to be lower than the long-term C-Q relationships when rainfall coincides with dry antecedent conditions. The C-Q relationships during rainfall on wet antecedent conditions were not significantly different from the long-term relationship. We argue that hydrologic connectivity to the nitrate sources, here represented by the values of event runoff coefficient, is crucial to explain deviations from the long-term C-Q relationship during different event types.

Finally, we found that flatter catchments with high denitrification potential (i.e., deep soils, presence of sedimentary aquifers), as well as catchments with agricultural areas located farther from the stream or with top-loaded nitrate profile, exhibit an enhanced nitrate transport during snow-impacted events and lower nitrate concentrations during events induced by rainfall with dry antecedent condition compared to the long-term C-Q relationship. Catchments with these characteristics are prone to generate disproportional loads during snow-impacted events, exacerbating ecological risk for receiving water

bodies. Findings from this study improve our understanding of the effects of runoff event types on nutrient dynamics and provide valuable insights for optimizing water quality management and monitoring.

*Data availability:*
Water quality dataset is available online at: http://dx.doi.org/10.4211/hs.a42addcbd59a466a9aa56472dfef8721.
Runoff event classification is available under request at: https://zenodo.org/record/3575024.
Catchment characteristics dataset is available at: https://doi.org/10.4211/hs.82f8094dd61e449a826afdef820a2c19.

*Author contributions*. FS contributed with formal analyses and writing. All the authors contributed in the conceptualization and writing.

*Competing interests*. The authors declare that they have no conflict of interest.

*Acknowledgements*. The financial support of Helmholtz Centre for Environmental Research – UFZ (DYNAMO Cohort). Larisa Tarasova was supported by the German Research Foundation ("Deutsche Forschungsgemeinschaft," DFG) in terms of the research group FOR 2416 "Space-Time Dynamics of Extreme Floods (SPATE)". Jana von Freyberg was supported by the Swiss National Science Foundation SNSF (grant PR00P2_185931). We thank Camile Minaudo and two anonymous reviewers for their valuable suggestions that helped to improve the original manuscript.

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

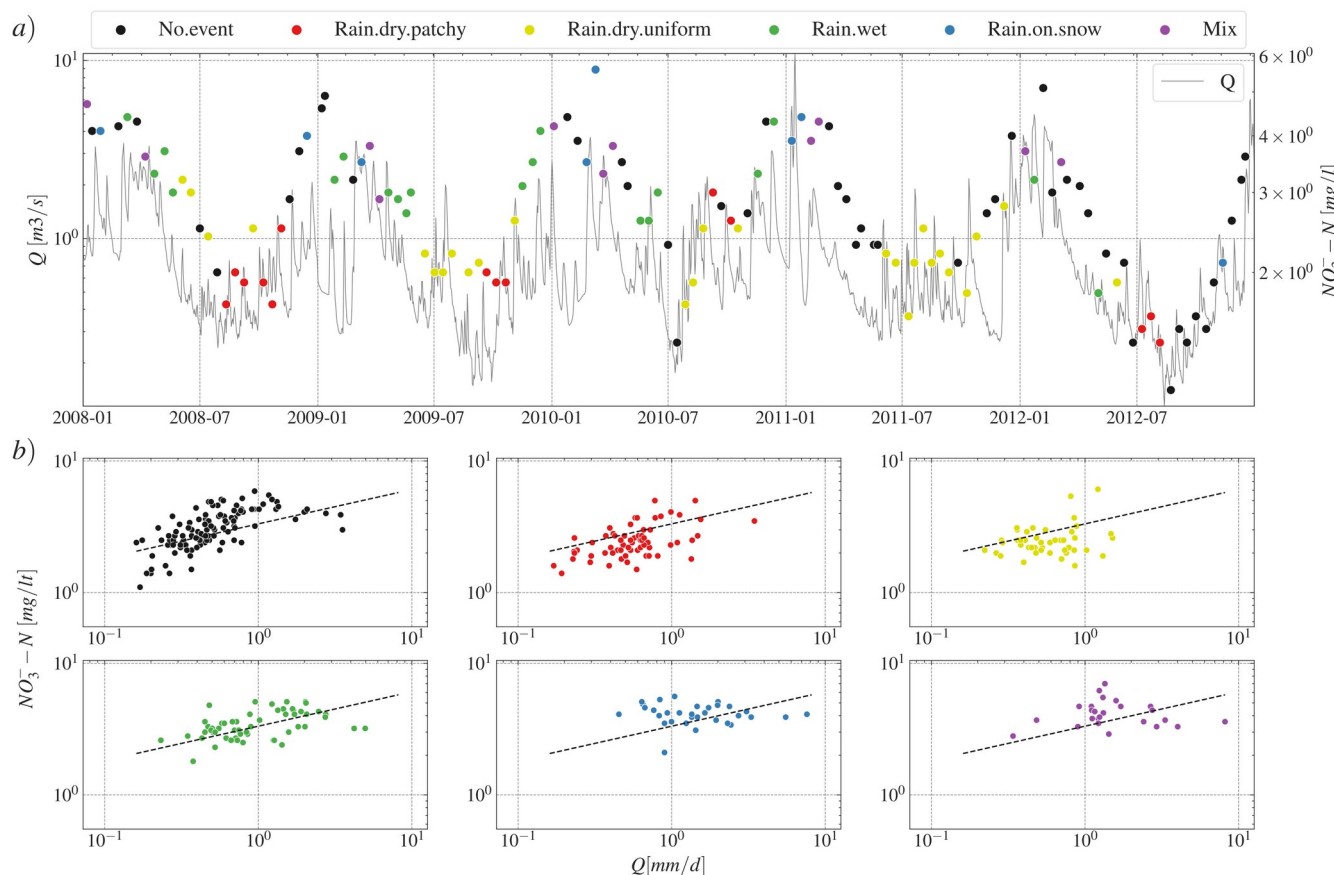

**Figure 1: a) Time series of daily discharge and bi-weekly grab sample nitrate concentration during event and no event conditions in the Naab River at the gauge of Unterkoeblitz, Bavaria over a period of five years. Event types are differentiated by colors (see Figure 3 for details). b) Double logarithmic plot of C-Q pairs for samples (from 2000 to 2012) taken during different event types and no event conditions. Black dashed lines show the long-term C-Q relationship (same line in each subplot) obtained from linear**

**regression in a double logarithmic plot of C-Q values for all available samples.**

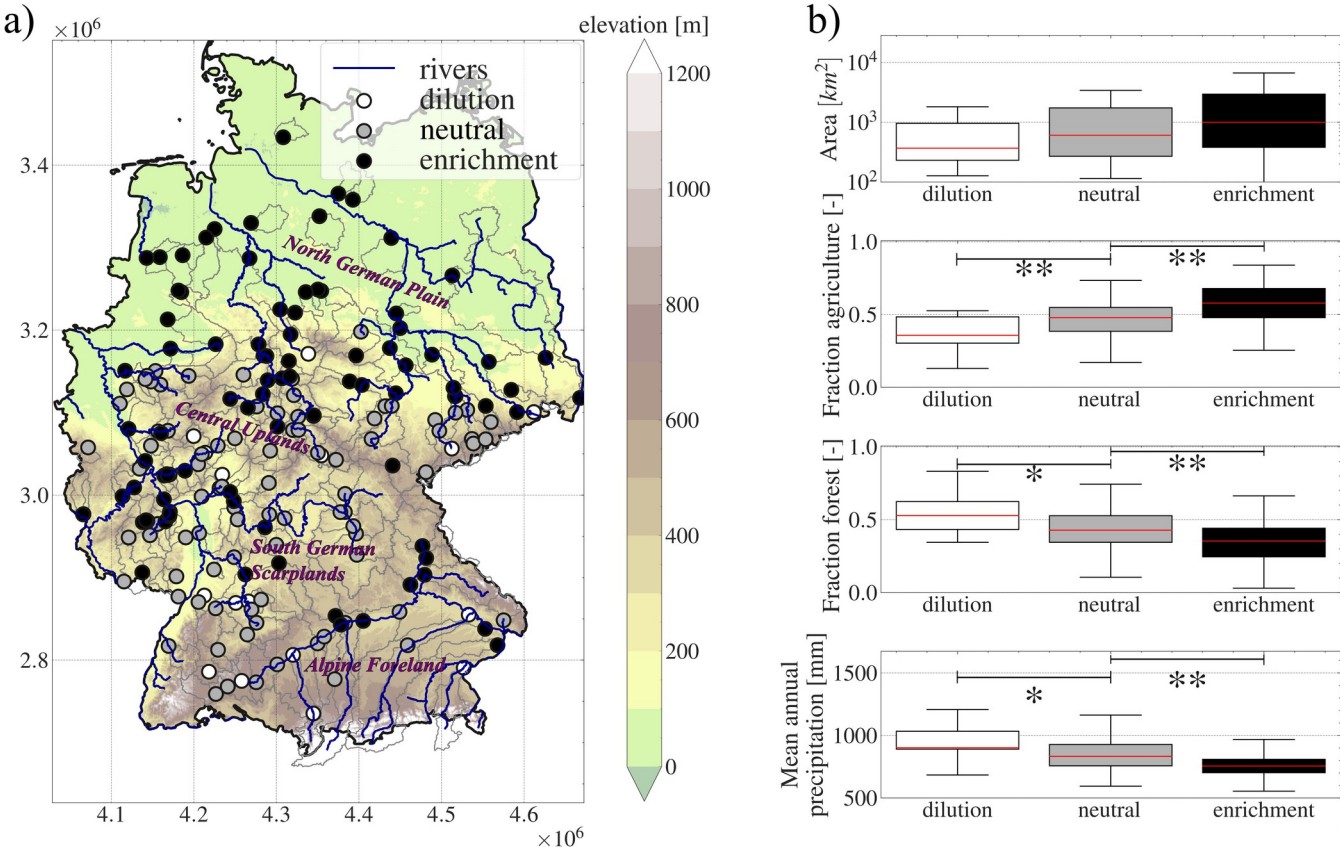

**Figure 2: a) Study area and stations of nitrate concentration measurements in stream water. Gray lines show catchment boundaries. Catchment outlets (points) are color-coded according to the long-term export pattern (dilution, neutral and enrichment). Blue lines show the main rivers. The background colormap corresponds to the elevation. Purple labels indicate German natural regions b) Area, fraction of agriculture, fraction of forest and mean annual precipitation of study catchments grouped according to export patterns (dilution, neutral and enrichment). Red lines show medians of boxplots and significance of median differences between adjacent boxplots was estimated using Kruskal-Wallis test (displayed as * for p<0.05 and ** for p<0.01).**

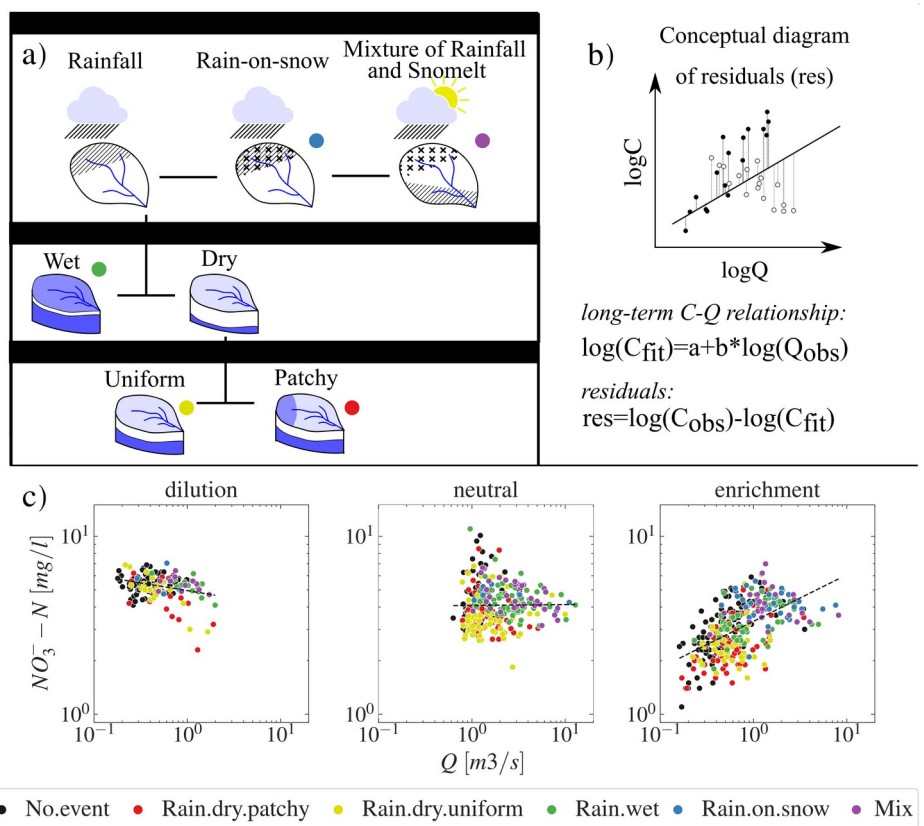

**Figure 3: a) Hierarchical scheme for event classification (modified from Tarasova et al., (2020); classification criteria are provided in Table S1). Colored dots located next to the five different event types indicate their markers. b) Cobs and Qobs are observed concentration and discharge, Cfit is the nitrate concentration estimated from fitting the long-term C-Q relationship with a linear relation in log-log space, and res is the residual value. c) C-Q plots for three different catchments attributed to different long-term nitrate export patterns based on the logC-logQ slope b, i.e., dilution (b<-0.1, the Würm River in Pforzheim), neutral (b~0, the Wupper River in Opladen) and enrichment (b>0.1, the Naab River in Unterkoeblitz).**

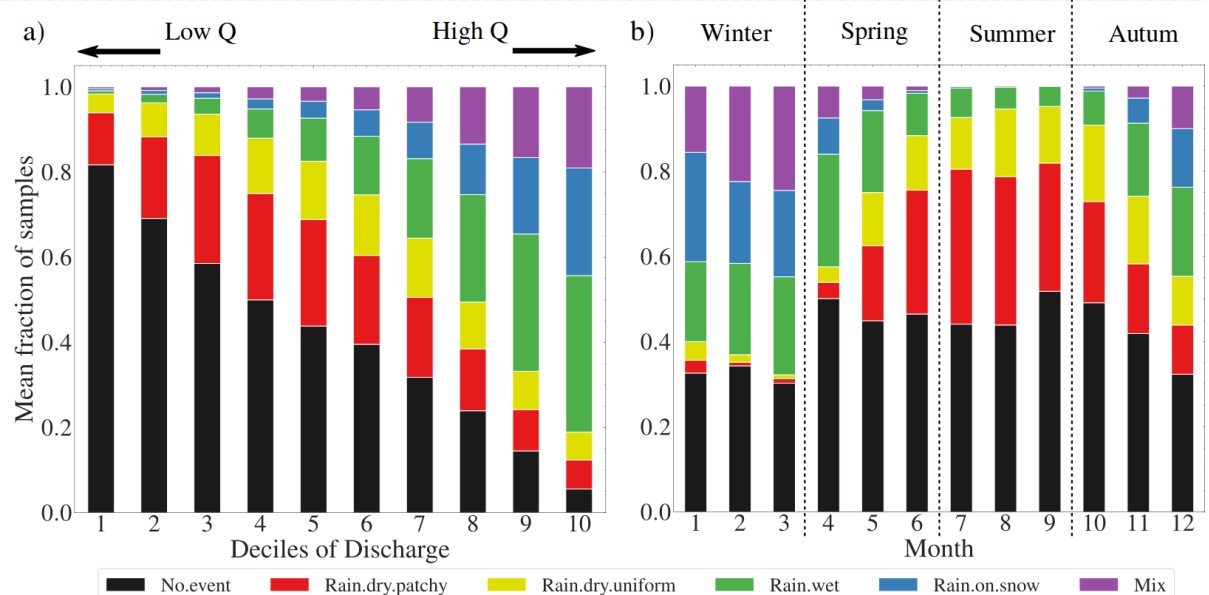

**Figure 4: a) Mean fraction of samples linked to each event type according to each catchment decile of discharge, and b) seasonal distribution of mean fraction of samples linked to each event type in the study catchments.**

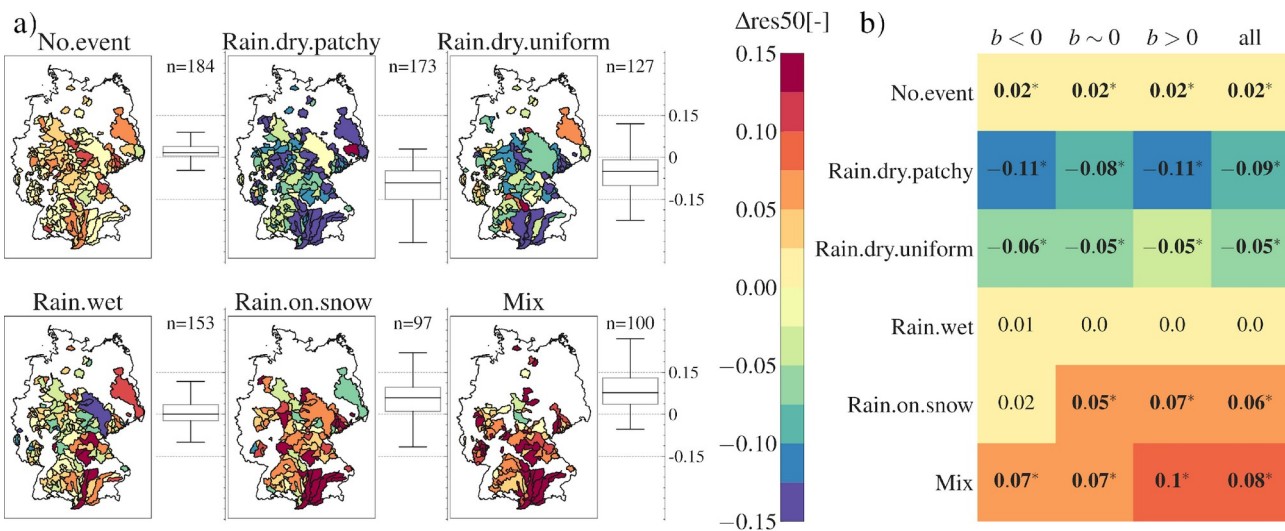

**Figure 5: Median deviations of nitrate concentrations from the long-term C-Q relationships (Δres50) a) Δres50 values of different event types for each catchment. On the right-hand side of each map, boxplots show the distribution of Δres50 values across catchments for each event type (box limits represent the interquartile range and whiskers correspond to the 5th and 95th percentiles). b) Heatmap of Δres50 values averaged across different groups of catchments, considering all nitrate data for each event type and No.event. The three first columns of the heatmap correspond to one of the long-term export patterns (i.e., dilution (slope b<0), neutral (slope b~0) and enrichment (slope b>0)) and the fourth column corresponds to all study catchments. Bold font and * indicates significant differences (Kruskal-Wallis test, p<0.05) between median deviations across catchments for each event type and median deviation across catchments of all nitrate samples.**

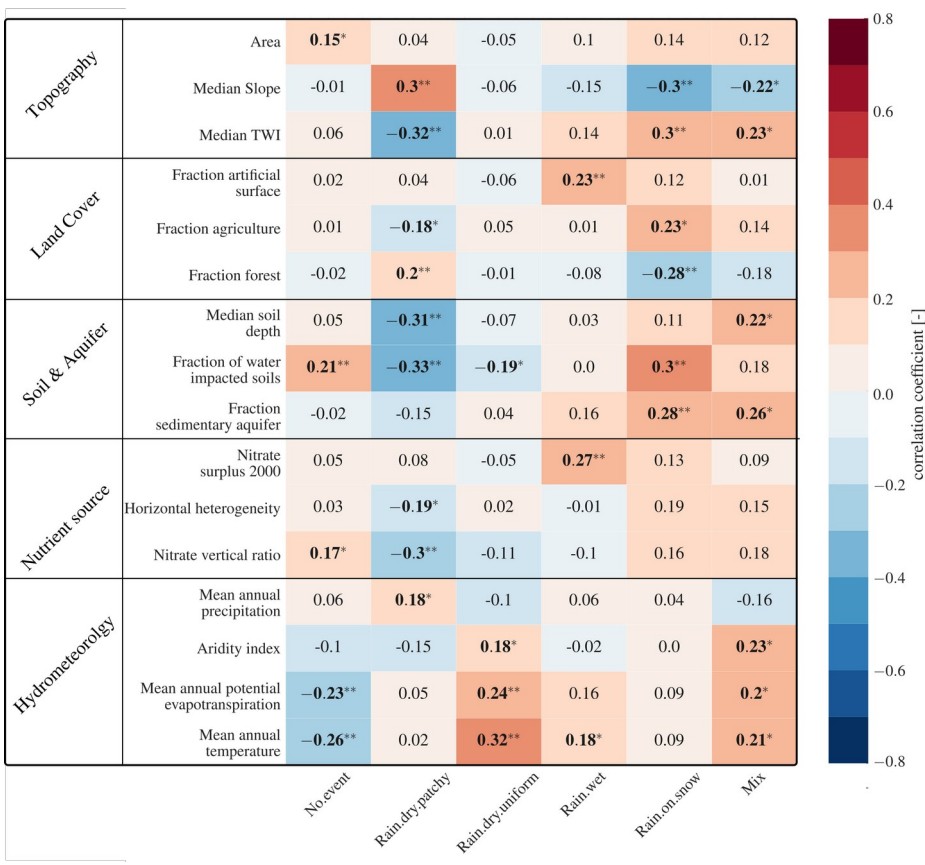

**Figure 6: Spearman rank correlation coefficient between deviations of nitrate concentrations from the long-term C-Q relationships (Δres50) of a particular event type across study catchments and catchment descriptors. Significant correlations are indicated by bold font and * for p<0.05 and ** for p<0.01.**

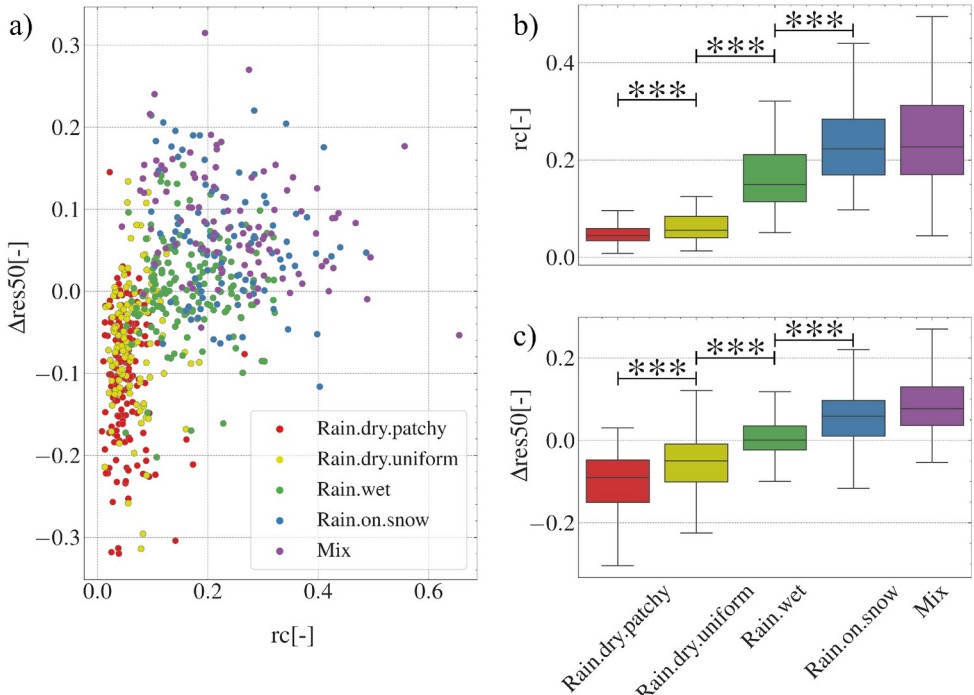

Figure 7: (a) Relationship between Δres50 for each catchment and event type and median runoff coefficient (rc); runoff coefficient is not defined for No.event. (b) Variability of runoff coefficients (rc) for each event type and (c) median residuals for each event type. Significance of median differences between adjacent boxplots was estimated using Kruskal-Wallis test (displayed as *** for p<0.001).