# Peer review of "Disentangling Scatter in Long-Term Concentration-Discharge Relationships: the Role of Event Types"

_EGUsphere, 2022_

## Author Comment (AC2)

**Manuscript: Disentangling Scatter in Long-Term Concentration-Discharge Relationships: the Role of Event Types**

Response to Anonymous Reviewer 2

We thank the reviewer for a comprehensive review. Below we provide our point-by-point replies (black color) to the reviewer comments (blue color). New or modified text in the revised manuscript is presented in italics.

1. The paper aims to provide explanation for deviations from long-term C-Q behaviour for different types of hydrological conditions. The authors claim that they are first in doing so, but the only novel thing in this study is a large number of catchments that are investigated. The discussion and implications are pretty much the same as in other studies by the research team, highlighting the incremental character of this study. Thus, to grant the publication of this paper, the authors need to convince the readers about novelty of their work, in light of recent publications in this field.

We apologize that the novelty of our study was not clearly highlighted in the manuscript. The main novelty of our study is in combining the hydrological event classification framework with long-term low-frequency data. To the best of our knowledge runoff event characteristics to explore nitrate dynamics were previously mostly considered in high-frequency studies across individual or few catchments (e.g. Bauwe *et al.*, 2015; Knapp *et al.*, 2020; Heathwaite and Bieroza, 2021), while only Minaudo et al. (2019) and Pohle et al., (2021) considered hydrological conditions using a large sample of catchments and low-frequency data. Combining the information about the hydrological events at the time of sampling with low-frequency data enabled to find systematic deviations in the long-term C-Q relationships induced by different hydrological conditions in a large sample of catchments. Moreover, our large dataset of catchments with contrasting characteristics allows for deducing mechanisms behind the spatial variability of nitrate C-Q deviations across German catchments. Finally, the abundance of low-frequency data worldwide and the transferable nature of the applied event classification framework paves the way to further applications in contrasting environments to better understand scatter in long-term C-Q relationships.

To clarify and highlight the novelty of the study we propose to add/modify the following lines of the manuscript.

Low-frequency data combined with runoff event classification in a large number of contrasting catchments

Abstract L18: *"This study combines a hydrological runoff event classification framework with low-frequency nitrate samples in 184 catchments to explore the role of different runoff events in shaping long-term C-Q relationships and their variability across contrasting catchments. "*

Introduction L48: *"The scatter of C-Q relationships might also be related to hydrologic conditions at the time of sampling (Knapp et al., 2020, Musolff et al., 2021), which are investigated for a large number of catchments only by a few recent studies (Minaudo et al., 2019; Pohle et al., 2021)."*

Spatial variability of C-Q deviations across contrasting catchments

Abstract L25: *"Using long-term, low-frequency nitrate data we demonstrate for the first time for a large set of catchments that runoff event types shape observed scatter in long-term C-Q relationships according to the level of hydrologic connectivity characteristic of each runoff event type. In addition, we hypothesize that the level of biogeochemical attenuation taking place in catchments can partially explain the spatial variability of the scatter during different event types."*

Conclusions L452: *"Moreover, we inferred using catchment descriptors physical mechanisms that possibly explain the spatial variability of this scatter."*

Systematic deviations of C-Q relationships linked to the runoff event types

Introduction L96: *"Our study aims for the first time to investigate the presence of systematic deviations in long-term C-Q relationships produced by different runoff event types in a large set of catchments."*

Results L271: *"We found systematic differences in the direction and magnitude of deviations of nitrate concentrations (∆res50) from the long-term C-Q relationships during different types of runoff events despite the large variety of study catchments (Fig. 5)."*

Transferability of our methods

Discussion L440: *The abundance of low-frequency data worldwide and the transferable nature of the applied event classification framework provide the means for further applications in contrasting environments to better understand the origins of scatter in long-term nitrate C-Q relationships.*

> 2. I understand that the authors want to show off the contributions from their own team, but there are plenty other papers, not published by your group, that you could refer to in your discussion.

We apologize if we have overlooked relevant references in our manuscript. To show contributions from a larger number of research groups, we will modify the cited papers by adding or replacing references. We propose the following changes:

L33: Moreover, due to long-lasting legacy effects a delay in reducing riverine nitrate concentration was reported in many catchments *(Tesoriero et al., 2013; Meter and Basu, 2017; Bieroza et al., 2018; Chang et al., 2021).*

L38: The shape of C-Q relationships encodes export patterns and reflects the temporally varying quantities of critical substances such as nutrients delivered to streams *(Godsey et al., 2009; Meybeck and Moatar, 2012; Rose et al., 2018).*

L43: Differences in long-term C-Q-relationships among catchments can be associated with differences in the availability and spatial distribution of solute sources *(Musolff et al., 2017; Dupas et al., 2019; Zhi et al., 2019; Casquin et al., 2021), t*heir hydrologic connectivity *(Seibert et al., 2009; Dupas et al., 2016; Covino, 2017)* and biogeochemical processes within the soil and stream that can retain or permanently remove nitrate from streamwater *(Mulholland et al., 2008; Dupas et al., 2016; Moatar et al., 2017; Benettin et al., 2020).*

L74: At seasonal scale nutrient transport to streams can be increased with higher hydrologic connectivity in catchments with abundant sources *(Martin et al., 2004; Veith et al., 2020; Guillemot et al., 2021).*

L285: We argue that during snow-impacted events hydrologic connectivity of sources is high due to elevated wetness conditions (Stieglitz et al., 2003) which is consistent with previously reported high nitrate concentration during the winter period *(Martin et al., 2004; Ocampo et al., 2006; Yang et al., 2018).*

L299: Rain.dry.uniform and Rain.dry.patchy events occur more often during the dry season when nitrate concentrations are reported to be lower in several studies *(House et al., 2001; Guillemot et al., 2021).*

L384: Deep sedimentary aquifers have a high potential for denitrification due to a great availability of electron donors, longer transit times, and more anoxic conditions due to sufficient reduction capacity (Kunkel et al., 2004; Wendland et al., 2008; Knoll et al., 2020) producing a lower nitrate supply in deeper soils compared to shallow soil *(Dupas et al., 2016).*

L400: *Many studies have highlighted the importance of agricultural sources for nitrate export patterns in several catchments (e.g., Moatar et al., 2017; Minaudo et al., 2019; Casquin et al., 2020; Weber et al., 2020).*

L423: *A reduction in the frequency of snow-impacted events was already shown in Germany over the last decades (Fontrodona Bach et al., 2018; Chan et al., 2020; Taszarek et al., 2020).*

Specific comments

> 3. Line 16 grammar

Thank you for this suggestion. We will modify the text.

L16: *"Although previous studies investigated the origins of this scatter in individual or in a few catchments, the role of different runoff event types across a large set of catchments is not yet fully understood."*

> 4. Line 16 how about Winter et al? This topic seems to have been already covered by your colleagues, so what is the novel aspect of this study? There have been also other paper studying how different storm event response contribute to scatter in C-Q data making this statement untrue, please update the list of previous studies on the topic in the introduction

There are in fact considerable differences between Winter et al. (2021) and this manuscript. The work of Winter et al. focuses on the variability between runoff events during a limited 4-years period considering only samples takan during runoff events and using high-frequency data in only 6 neighboring catchments in Central Germany. The study finds that variability of hysteresis patterns decreases from runoff events induced by rainfall with dry antecedent conditions to snow-impacted events. In contrast, this manuscript uses low-frequency across 184 catchments data and investigates the effect of runoff event types in long-term C-Q relationships. The increase of nitrate concentration during snow-impacted and the decrease decrease during rainfall events with dry antecedent conditions of Winter et al. (2021) is also confirmed in our work. However, a much larger number of catchments with contrasting characteristics used in this study allow us to investigate systematic nitrate deviations from long-term C-Q relationships across catchments and attribute spatial patterns of deviations to potential physical mechanisms using catchment characteristics. We will modify L16 mentioned by the referee to clarify the differences between the two studies.

L16: *"Although previous studies investigated the origins of this scatter in individual or in a few catchments, the role of different runoff event types across a large set of catchments is not yet fully understood."*

We apologize if we have overlooked relevant papers on the scatter in C-Q relationships due to the different storm responses. We have added now additional references to the Introduction.

L53: The cause of this scatter can also be traced to a variety of responses observed at the event-scale in several studies with high-frequency data in single or a few catchments *(e.g., Bowes et al., 2015; Lloyd et al., 2016; Koenig et al., 2017; Gorski and Zimmer, 2021).*

L57: Disparate patterns of the event C-Q relationships in a catchment over time are mainly attributed to varying dominant flow sources (e.g., groundwater, shallow subsurface flow), antecedent wetness conditions (Inamdar et al., 2006; Vaughan et al., 2017; Knapp et al., 2020), *time of fertilizer application (Bowes et al., 2015; Dupas et al., 2016; Outram et al., 2016), biogeochemical cycling (Heathwaite and Bieroza, 2021)* and runoff event characteristics or types *(Butturini et al., 2006; Bauwe et al., 2015; Chen et al., 2020; Knapp et al., 2020; Heathwaite and Bieroza, 2021).*

> 1. Line 22 'indicating low nitrate concentrations' – this does not make sense

Thank you for pointing this out. We will modify the text to clarify this issue:

*"In contrast, negative deviations occur mostly for rainfall-induced events with dry antecedent conditions, indicating the occurrence of lower nitrate concentrations in river flows than their long-term pattern values during this type of events."*

> 5. It is not clear if you analyse high-frequency or low-frequency C-Q data, this should be clarified at the very beginning of the paper. Without this information it is difficult to judge the quality of your hypotheses.

Thank you for the comment. We agree that this should be clarified s in the Introduction section. We will add the following sentence in line 56:

*"Our study relies on low-frequency nitrate data, which is often used to build long-term C-Q relationships (e.g. Cartwright et al., 2020, Diamond and Cohen 2018). However, studies with high-frequency data*

*found large variability in the C-Q patterns during events (e.g. Knapp et al., 2020; Dupas et al., 2016; Vaughan et al., 2017) that might add scatter to the long-term C-Q relationship.*

6. Figure 1 should be part of methods or results but not introduction

Thank you for this suggestion. We will move the figure to the Methods section.

7. Hypothesis 1 is not clear. Do you mean individual C-Q points?

Thank you for pointing this out. To clarify this we will modify it as follows:

*"1 Do samples collected during different event types deviate differently from the long-term C-Q relationships observed at the catchment outlets?"*

8. Not clear how daily discharge data can provide information about short storm events with duration of hours?

Thank you for your question. We use daily streamflow to identify events. This implies that the shortest event that can be captured has a duration of at least 1 day. Any event shorter than 1 day cannot be captured with the available data. We will modify the following sentence in Line #135 for clarification:

*"The method includes baseflow separation, precipitation attribution and an iterative procedure to adjust site-specific thresholds for the refinement of multi-peak events. We use daily streamflow data to identify events. This implies that only events longer than 1 day are captured."*

9. In this sense, using a term 'event classification' is misleading. I would rather use classification of 'hydrological conditions'.

We prefer to keep the term "event classification" instead of "hydrological conditions" as the former more accurately represents the information combined in the event types and is a standard in the hydrological literature (e.g., Bauwe et al., 2015; Chen et al., 2020; Ross et al., 2019; Xie et al., 2019). Apart from information on hydrological conditions often used (i.e., wetness conditions) it also includes information on the nature of precipitation events and spatial distribution of soil moisture.

10. Since you have low-frequency samples they are sampled randomly over the hydrograph. So samples that belong to the same hydrological condition can have been sampled on a rising, falling limb of the hydrograph or baseflow conditions. Thus, some of your scatter in each hydrological condition group can be attributed to when on the hydrograph your samples were taken.                                                    Please                                        clarify.

I have just noticed that Reviewer 1 expressed similar concerns regarding the role of C-Q hysteresis. This is a key weakness of your approach.

We appreciate the reviewer's comment on the possible effect of hysteresis at the event scale. We agree that this effect requires additional attention in the manuscript. The first reviewer have raised a similar concern, therefore below we repeat our response to reviewer 1 (Comment 1).

We quantified the proportion of samples taken in the rising limb, falling limb and near to the discharge peak (near-to-peak) of the event hydrographs. The rising limb starts at the beginning of the runoff event and finishs one day before the day of the peak discharge. The falling limb starts one day after the day of the peak discharge and finishs at the end of the runoff event. The beginning and the end of the runoff events are obtained from the runoff event detection method explained in detail in the original manuscript (Lines 132-135). We defined near-to-peak as samples collected from one day before to one day after the day of the peak discharge. We allowed some overlap between near-to-peak and other two groups to use a larger number of samples than considering samples collected on the day of the peak of discharge only. Of the total samples taken during runoff event types 34% correspond to the rising limb, 55% to the falling limb and 30% to near-to-peak (11% of the samples were collected during the day of the peak discharge). This information will be shown in Figure S6a in the revised manuscript. In addition, we quantified the

deviations of the long-term C-Q relationship (Δres50) for samples taken during the rising limb, falling limb and near-to-peak. We computed the deviations for these three groups of samples following the same bootstrapping procedure shown in the Method section (Lines 165-172) of the original manuscript.

The new results provided in Figure S6b show that the deviations from the long-term C-Q relationships for different event types are very similar for all three cases (samples taken during falling limb, rising limb or near the event peak) and resemble the deviations that we have previously observed for all collected samples (Figure 5 in the main manuscript). This suggests that the relative time of sampling during an event does not affect deviations from the long-term C-Q relationships that we detected for different event types.

[Figure]

*Figure S6. a) Number of samples per catchment per event type corresponding to the samples taken during the rising limb, falling limb, or near to the peak (i.e., samples taken from one day before to one day after the peak of the hydrograph). b) Median deviations of nitrate concentrations from the long-term C-Q relationships (Δres50) for samples taken during the rising limb, falling limb, and near to the peak. Deviations are computed analogously as for Fig. 5 in the main manuscript. The three first columns of the heatmap correspond to one of the long-term export patterns (i.e., dilution (slope b<0), neutral (slope b~0), and enrichment (slope b>0)), and the fourth column corresponds to all study catchments. Bold font and * indicate significant differences (Kruskal-Wallis test, p<0.05) between median deviations across catchments for each event type and median deviation across catchments of all nitrate samples. At least 5 catchments with sufficient data (more than 10 samples per event type) are required to evaluate the significance of the deviations. Gray squares indicate cases where this requirement is not met.*

We will insert the following description in the revised manuscript in the Method section.

L181: *"Low frequency datasets such as the one used in our study might contain samples collected during different phases of the event hydrograph (e.g., falling or rising limb). This might hamper the interpretability of the results due to possible bias in observed nitrate concentration linked to the time of sampling and the hysteresis effect revealed in high-frequency observations (e.g., Lloyd et al., 2016; Vaughan et al., 2017). In fact, Pohle et al. (2021) showed systematic differences in nitrate concentration between samples collected during rising and falling limbs for numerous catchments in Scotland. To understand the potential effect of the hysteresis on the deviations from long-term C-Q (Δres50) we repeat the bootstrapping procedure described above considering samples collected during the rising limb, falling limb and near the event peak (near-to-peak). The rising limb of a runoff event starts at the beginning of*

*the event and finishes one day before the day of the peak discharge. The falling limb starts one day after the day of the peak discharge and finishes at the end of the runoff event. Moreover, we defined near-to-peak as samples collected from one day before to one day after the day of the peak discharge. Of the total samples taken during runoff event types 34% correspond to the rising limb, 55% to the falling limb and 30% to near-to-peak. Notice that definition of near-to-peak samples allows some overlap with the other two groups of samples to use a more balanced number of samples than considering samples collected on the day of the peak of discharge only (only 11% of the samples were collected during the day of the peak discharge). "*

We will add the following lines in the Result section.

L236: *The time of sampling in runoff events did not interfere with our main results (Fig. S6b). Although some data limitations for some group of samples (gray tiles in Fig. S6b), we could reproduce the analysis for most of the cases. We found that similarly to our results using all the samples (Fig. 5b) values of $\Delta res50$ for samples taken during the rising limb, near to the peak and falling limb, are positive for Rain.on.snow and Mix events, negative for Rain.dry.patchy and Rain.dry.uniform, and intermediate for Rain.wet events.*

We will add the following text discussing the results of the additional experiment to the Discussion section of the original manuscript.

L443: *Although the presence of the hysteresis effect might considerably affect nitrate concentration during rising and falling limbs of the event hydrograph in some catchments (Pohle et al, 2021) we found a similar direction of deviations from the long-term C-Q relationships when we considered samples taken during rising limb, falling limb and near to the peak (Fig S6b). Hence, our results suggest that the variability potentially added by the presence of hysteresis patterns is lower than the deviations observed for different event types from the long-term C-Q relationship. Increasing availability of high-frequency datasets coupled with new statistical modeling approaches might be used in the future to evaluate hysteresis-related effects in the existing long-term C-Q datasets to further disentangle inter- and intra-event variability of nitrate dynamics at larger scales"*

[revised manuscript text omitted]

Pohle I, Baggaley N, Palarea-Albaladejo J, Stutter M, Glendell M. 2021. A Framework for Assessing Concentration-Discharge Catchment Behavior From Low-Frequency Water Quality Data. *Water Resources Research* **57** (9): e2021WR029692 DOI: 10.1029/2021WR029692

Rose LA, Karwan DL, Godsey SE. 2018. Concentration–discharge relationships describe solute and sediment mobilization, reaction, and transport at event and longer timescales. *Hydrological Processes* **32** (18): 2829–2844 DOI: 10.1002/hyp.13235

Ross, C. A., Ali, G., Spence, C., Oswald, C., & Casson, N. (2019). Comparison of event-specific rainfall–runoff responses and their controls in contrasting geographic areas. Hydrological Processes, 33(14), 1961–1979. https://doi.org/10.1002/hyp.13460

Seibert J, Grabs T, Köhler S, Laudon H, Winterdahl M, Bishop K. 2009. Linking soil- and stream-water chemistry based on a Riparian Flow-Concentration Integration Model. *Hydrology and Earth System Sciences* **13** (12): 2287–2297 DOI: 10.5194/hess-13-2287-2009

Stieglitz M, Shaman J, McNamara J, Engel V, Shanley J, Kling GW. 2003. An approach to understanding hydrologic connectivity on the hillslope and the implications for nutrient transport. *Global Biogeochemical Cycles* **17** (4) DOI: 10.1029/2003GB002041

Taszarek M, Kendzierski S, Pilguj N. 2020. Hazardous weather affecting European airports: Climatological estimates of situations with limited visibility, thunderstorm, low-level wind shear and snowfall from ERA5. *Weather and Climate Extremes* **28**: 100243 DOI: 10.1016/j.wace.2020.100243

Tesoriero AJ, Duff JH, Saad DA, Spahr NE, Wolock DM. 2013. Vulnerability of Streams to Legacy Nitrate Sources. *Environmental Science & Technology* **47** (8): 3623–3629 DOI: 10.1021/es305026x

Vaughan MCH, Bowden WB, Shanley JB, Vermilyea A, Sleeper R, Gold AJ, Pradhanang SM,

Inamdar SP, Levia DF, Andres AS, et al. 2017. High-frequency dissolved organic carbon and nitrate measurements reveal differences in storm hysteresis and loading in relation to land cover and seasonality. *Water Resources Research* **53** (7): 5345–5363 DOI: 10.1002/2017WR020491

Veith TL, Preisendanz HE, Elkin KR. 2020. Characterizing transport of natural and anthropogenic constituents in a long-term agricultural watershed in the northeastern United States. *Journal of Soil and Water Conservation* **75** (3): 319–329 DOI: 10.2489/jswc.75.3.319

Weber G, Honecker U, Kubiniok J. 2020. Nitrate dynamics in springs and headwater streams with agricultural catchments in southwestern Germany. *Science of The Total Environment* **722**: 137858 DOI: 10.1016/j.scitotenv.2020.137858

Wendland F, Blum A, Coetsiers M, Gorova R, Griffioen J, Grima J, Hinsby K, Kunkel R, Marandi A, Melo T, et al. 2008. European aquifer typology: a practical framework for an overview of major groundwater composition at European scale. *Environmental Geology* **55** (1): 77–85 DOI: 10.1007/s00254-007-0966-5

Xie, H., Dong, J., Shen, Z., Chen, L., Lai, X., Qiu, J., Wei, G., Peng, Y., & Chen, X. (2019). Intra- and inter-event characteristics and controlling factors of agricultural nonpoint source pollution under different types of rainfall-runoff events. CATENA, 182, 104105. https://doi.org/10.1016/j.catena.2019.104105

Yang J, Heidbüchel I, Musolff A, Reinstorf F, Fleckenstein JH. 2018. Exploring the Dynamics of Transit Times and Subsurface Mixing in a Small Agricultural Catchment. *Water Resources Research* **54** (3): 2317–2335 DOI: 10.1002/2017WR021896

Zhi W, Li L, Dong W, Brown W, Kaye J, Steefel C, Williams KH. 2019. Distinct Source Water Chemistry Shapes Contrasting Concentration-Discharge Patterns. *Water Resources Research* **55** (5): 4233–4251 DOI: 10.1029/2018WR024257

---

## Author Response (AR1)

Manuscript Egusphere-2022-205

**Disentangling Scatter in Long-Term Concentration-Discharge Relationships: the Role of Event Types**

We would like to aknowledge the Editor and two Anonymous Reviers for providing detailed and constructive comments to our manuscript, which helped significantly to improve our original manuscript. This document provides our point-by-point replies (black color) to the reviewer or editor comments (blue color). New or modified text in the revised manuscript is presented in italics. In addition, we encolsed a revised manuscript with track changes in our response.

**Reply to Editor comments**

Two referees have evaluated your revised manuscript, and they raised minor to moderate/major issues that need to be addressed. Upon reading referee reports, I was going to suggest that you pay special attention to 1) referee 2's request that you better articulate the novelty of your work by discussing recent studies on c-Q relations, and 2) referee 1's suggestion that you discuss the impact of sampling timing on your results (potential sampling bias). I am pleased to see, from the Interactive Discussion page, that you have already taken steps to address those referee comments and others. I therefore return your manuscript for moderate revision. Upon reception, your revised manuscript and response document will be sent out for another round of review.

Thank you very much for handling our manuscript. In the following lines, we address all the comments of the reviewers.

**Reply to Reviewer #1**

General Comments:

1. This study presents a unique, spatially and temporally extensive dataset of nitrate C-Q relationships across 184 German catchments of varying size and land cover/land use from 2000-2015. The authors found that the degree of catchment hydrologic connectivity (and the closely related factor of runoff event type) strongly regulates the pattern of catchment nitrate export. Divergence of event-scale nitrate C-Q responses from the more generalized long-term response were attributed to a combination of catchment topographic properties and event type. The study dataset is impressive, providing a long-term view of catchment nitrate responses across gradients of event type, topography, and land use. The paper is very well written, making it both easy and enjoyable to read; there are only a few places in the manuscript where grammatical clarifications are needed. The statistical analyses presented in the paper are well-presented and wholly appropriate for the research questions that are being asked. Overall, this paper represents a meaningful addition to the existing C-Q literature and I recommend it for publication with only a few minor revisions. My main concern with the paper is that the potential influence of C-Q hysteresis on the observed patterns is not addressed anywhere in the paper. For example, if a particular catchment (or a particular event type) is characterized by a strong hysteresis signal, then the observed C-Q pattern (i.e., dilution or enrichment) would be highly influenced by the timing of sample collection. If a strong sampling bias exists where samples are more frequently collected on the rising limb relative to the falling limb (or vice versa), then the observed catchment or event C-Q signal might be confounded by the presence of hysteresis processes. I would not expect this to be

an issue for the long-term C-Q pattern, but it may be an issue for the event-scale patterns and this would, in turn, cause a problem for the interpretation of the "Δres50" term presented in the paper. Given the low frequency of sample collection in this dataset (biweekly to monthly), it seems likely that a particular runoff event would be represented in the dataset by a sample collected either on the rising limb OR the falling limb but not both. It would be fascinating to see an additional analysis of this extensive dataset that incorporates the potential influence of sample timing on the hydrograph, but this is likely beyond the scope of this paper in its current form. However, I do think the authors need to include at least some discussion of the potential influence of this potential "hysteresis effect bias" associated with low-frequency event sampling.

We appreciate the reviewer's comments and the ideas on the possible effect of hysteresis at the event scale. We agree that this effect requires additional attention in the manuscript.

We quantified the proportion of samples taken in the rising limb, falling limb and near to the discharge peak (near-to-peak) of the event hydrographs. The rising limb starts at the beginning of the runoff event and finishs one day before the day of the peak discharge. The falling limb starts one day after the day of the peak discharge and finishs at the end of the runoff event. The beginning and the end of the runoff events are obtained from the runoff event detection method explained in detail in the Methods section (Lines 152-157 in the revised manuscript). We defined near-to-peak as samples collected from one day before to one day after the day of the peak discharge. We allowed some overlap between near-to-peak and other two groups to use a larger number of samples than considering samples collected on the day of the peak of discharge only. Of the total samples taken during runoff event types 34% correspond to the rising limb, 55% to the falling limb and 30% to near-to-peak (11% of the samples were collected during the day of the peak discharge). This information will be shown in Figure S6a in the revised manuscript. In addition, we quantified the deviations of the long-term C-Q relationship (Δres50) for samples taken during the rising limb, falling limb and near-to-peak. We computed the deviations for these three groups of samples following the same bootstrapping procedure shown in the Method section (Lines 186-202 in the revised manuscript).

The new results provided in Figure S6b show that the deviations from the long-term C-Q relationships for different event types are very similar for all three cases (samples taken during falling limb, rising limb or near the event peak) and resemble the deviations that we have previously observed for all collected samples (Figure 5 in the main manuscript). This suggests that the relative time of sampling during an event does not affect deviations from the long-term C-Q relationships that we detected for different event types.

[Figure]

*Figure S6. a) Number of samples per catchment per event type corresponding to the samples taken during the rising limb, falling limb, or near to the peak (i.e., samples taken from one day before to one day after the peak of the hydrograph). b) Median deviations of nitrate concentrations from the long-term C-Q relationships (Δres50) for samples taken during the rising limb, falling limb, and near to the peak. Deviations are computed analogously as for Fig. 5 in the main manuscript. The three first columns of the heatmap correspond to one of the long-term export patterns (i.e., dilution (slope b<0), neutral (slope b~0), and enrichment (slope b>0)), and the fourth column corresponds to all study catchments. Bold font and * indicate significant differences (Kruskal-Wallis test, p<0.05) between median deviations across catchments for each event type and median deviation across catchments of all nitrate samples. At least 5 catchments with sufficient data (more than 10 samples per event type) are required to evaluate the significance of the deviations. Gray squares indicate cases where this requirement is not met.*

We inserted the following description in the revised manuscript in the Method section.

L204 revised manuscript: *"Low frequency datasets such as the one used in our study might contain samples collected during different phases of the event hydrograph (e.g., falling or rising limb). This might hamper the interpretability of the results due to possible bias in observed nitrate concentration linked to the time of sampling and the hysteresis effect revealed in high-frequency observations (e.g., Lloyd et al., 2016; Vaughan et al., 2017). In fact, Pohle et al. (2021) showed systematic differences in nitrate concentration between samples collected during rising and falling limbs for numerous catchments in Scotland. To understand the potential effect of the hysteresis on the deviations from long-term C-Q (Δres50) we repeat the bootstrapping procedure described above considering samples collected during the rising limb, falling limb and near the event peak (near-to-peak). The rising limb of a runoff event starts at the beginning of the event and finishes one day before the day of the peak discharge. The falling limb starts one day after the day of the peak discharge and finishes at the end of the runoff event. In addition, we defined near-to-peak as samples collected from one day before to one day after the day of the peak discharge. Of the total samples taken during runoff event types 34% correspond to the rising limb, 55% to the falling limb and 30% to near-to-peak. Notice that definition of near-to-peak samples allows some overlap with the other two groups of samples to use a more balanced number of samples than considering samples collected on the day of the peak of discharge*

*only (11% of the samples were collected during the day of the peak discharge). "*

We added the following lines in the Result section.

L277 revised manuscript: *"The time of sampling in runoff events did not interfere with our main results (Fig. S6b). Although data limitations for a few groups of samples (gray tiles in Fig. S6b), we could reproduce the analysis for most of the cases. We found that similarly to our results using all the samples (Fig. 5b) values of ∆res50 for samples taken during the rising limb, near to the peak and falling limb, are positive for Rain.on.snow and Mix events, negative for Rain.dry.patchy and Rain.dry.uniform, and intermediate for Rain.wet events."*

We added the following text discussing the results of the additional experiment to the Discussion section of the revised manuscript.

L506 revised manuscript: *Although the presence of the event-scale hysteresis effect might considerably affect nitrate concentration during rising and falling limbs of the event hydrograph in some catchments (Pohle et al, 2021) we found a similar direction of deviations from the long-term C-Q relationships when we considered samples taken during rising limb, falling limb and near to the peak (Fig S6b). Hence, our results suggest that the variability potentially added by the presence of hysteresis patterns is lower than the deviations observed for different event types from the long-term C-Q relationship. Increasing availability of high-frequency datasets coupled with new statistical modeling approaches might be used in the future to evaluate hysteresis-related effects in the existing long-term C-Q datasets to further disentangle inter- and intra-event variability of nitrate dynamics at larger scales."*

Specific Comments:

2. Introduction: Very well-written and cited, providing a concise but informative review of the relevant C-Q literature. However, the Introduction focuses heavily (almost exclusively) on the hydrologic drivers of observed C-Q patterns, with little mention of the role of biogeochemical drivers. Particularly in the case of nitrate, biogeochemical drivers—emphasizing the "bio" aspect-- can also influence C-Q patterns. Because this paper focuses solely on nitrate concentrations, I think it is worth mentioning the potential role of biogeochemical processes as drivers of the observed C-Q patterns (this might fit well in the paragraph starting on L56 or after). For example, seasonality of microbial processes might influence soil nitrate concentrations and affect the observed patterns of C-Q especially during seasonal events (e.g., rain-on-snow). Similarly, one might expect the "C" side of the nitrate C-Q relationship to be strongly influenced by the timing of nitrogen fertilizer applications in agricultural catchments. In each of these two examples, the biogeochemical drivers exert as much (of not more) control on the C-Q relationship as the hydrologic drivers. Salli Thompson's 2011 paper "Relative dominance of hydrologic versus biogeochemical factors on solute export across impact gradients" might be a useful paper to consider here.

We thank the reviewer for the point made here on the role of biogeochemical processes. We agree that this point was not addressed sufficiently in the Introduction. We revised accordingly and we added the following lines to the introduction section.

L49 revised manuscript: "*Biogeochemical processes that affect nutrient cycles in soil and water might add variability to long-term C-Q relationships. The effectiveness of the denitrification process, which removes nitrate from the soil, depends on periodic environmental factors such as temperature and soil moisture and the availability of electron donors (Korom et al., 2012; Ortmeyer et al., 2021). Instream removal processes are also more efficient during low flows and higher temperatures, adding more variability to the low-flow portion of the long-term C-Q relationships (Dehaspe et al., 2021; Moatar et al., 2017). Moreover, the availability of nitrate sources is balanced by fertilizer application and*

*mineralization of organic nitrogen compounds and hence varies in time adding temporal variability to C-Q relationships. The time of fertilizer application is often unknown, and the mineralization processes depend on chemical soil conditions and environmental factors (e.g., soil moisture and temperature) that mediate communities of microorganisms (Curtin et al., 2012; Guntiñas et al., 2012). On the other hand, average residence times of nitrate in agricultural catchments can last for decades, producing a legacy in soil (Meter et al., 2016; Puckett et al., 2011; Tesoriero et al., 2013; Vervloet et al., 2018) that can buffer the periodic effect of biogeochemical processes reducing the variability in the concentration of nitrate (Basu et al., 2011; Bieroza et al., 2018; Thompson et al., 2011).*

L74 revised manuscript: *"Disparate patterns of the event C-Q relationships in a catchment over time are mainly attributed to varying dominant flow sources (e.g., groundwater, shallow subsurface flow), antecedent wetness conditions (Inamdar et al., 2006; Knapp et al., 2020; Vaughan et al., 2017), time of fertilizer application (Bowes et al., 2015; Dupas et al., 2016; Outram et al., 2016), biogeochemical cycling (Heathwaite and Bieroza, 2021) and runoff event characteristics or types (Butturini et al., 2006; Bauwe et al., 2015; Chen et al., 2020; Knapp et al., 2020)."*

3. Methods: If it is possible with your dataset to quantify the proportion of rising limb and falling limb samples, it would be good to include that quantitative information in the Methods section. If the proportions of the two are widely unbalanced, then the potential influence of that sampling bias on your results should be discussed in the Discussion. If it is not possible to determine the rising- or falling-limb status of samples in your dataset, then a brief acknowledgement of the implications of this should still be included in the Methods.

Thanks for the suggestion, we agree that this is important to show. We now include this information in the Methods section and add Figure S6. Please refer to Comment 1 for more details.

4. L134: What is meant here by "precipitation attribution"? Does this mean precipitation classification as rain or snow? Otherwise, I'm not sure what precipitation would be attributed to.

By the term precipitation attribution we mean that the method links runoff events with the corresponding inducing rainfall and/or snowmelt event. We clarified this in the revised manuscript:

L154 revised manuscript: "The method includes baseflow separation, precipitation event attribution *(i.e., corresponding inducing rainfall and/or snowmelt events are linked to runoff events)* and an iterative procedure to adjust site-specific thresholds for the refinement of multi-peak events."

5. L243-247: For these correlation analyses, how did you account for the potential interaction between catchment topographic characteristics and land use? For example, one would expect at least some of the flatter catchments to also be used for agriculture (indeed, Figure S4 seems to indicate this). Thus, a simple correlation between median catchment slope and nitrate C-Q response is not straightforward if it does not somehow control for potential biases due to land use effects on nitrate availability.

We agree with the reviewer that there could be considerable intercorrelation between the characteristics as we acknowledge in the original manuscript (L303 revised manuscript) for the fraction of forest. We additionally highlighted this point in the Results and Discussion section of the revised manuscript:

L291 revised manuscript: "*Specifically, flatter catchments (low median topographic slope) with greater soil depths that are mostly located in the Northern Germany and Alpine Foreland tend to exhibit more positive residuals for Rain.wet, Rain.on.snow and Mix events, and more negative residuals for Rain.dry.patchy events and samples taken during no event conditions (Fig. 5a). Catchments with these characteristics often also have a higher fraction of agricultural land cover (Fig. S4); however the latter feature shows less significant correlations with nitrate C-Q deviations.*"

L440 revised manuscript: "*We acknowledge that catchment characteristics might be highly correlated (Fig. S4). Flatter catchments often exhibit higher fractions of agriculture, therefore more diffuse source availability. Although the correlation of the fraction of agriculture and C-Q deviations during Rain.on.snow events was less significant than topographic descriptors, a potential increment of diffuse sources in flatter catchments might also enhance the mechanism of nitrate bypassing the buffer capacity of catchments during Rain.on.snow events generating higher C-Q deviations.*"

6. L253: "Instead, we *observed* strong..."? It seems like a word is missing here…

Revised as suggested, please refer to line 301 in the revised manuscript.

7. L264-266: It would provide useful context here to also provide the ranges around these median values, not only the medians themselves.

We appreciate the reviewer's insightful suggestion. We added the ranges for the coefficient of variation across event types in the revised manuscript. We report the coefficient of variation of median runoff coefficient across catchments since mean values of Rain.dry.patchy and Rain.dry.uniform events are very close to zero and obscure computation of their coefficient of variations.

L312 revised manuscript: "*Event runoff coefficients exhibit a larger variability across event types than across catchments for most of the catchments. Catchment median event runoff coefficients exhibit a coefficient of variation of 41% across catchments. Nevertheless, median runoff coefficients of event types exhibit coefficients of variation in different catchments from 12% to 118%, with a median value of 67% across catchment.*"

8. L276-296: These two paragraphs are basically invoking the same hydrologic driver for the observed C-Q patterns: catchment wetness status associated with a given event type. But catchment wetness also changes *during* events, and this is where the need to consider potential hysteresis effects / sampling biases becomes important. I am not sure where a discussion of this issue fits in best in the Discussion section, but it should be included somewhere.

Thanks for the insightful comment. We include this concern in the new paragraphs added to the discussion section. Please refer to our response to Comment 1.

9. L358: The word "wetness" is not needed here.

We agree with the comment and we will remove the word. Please refer to line 408 in the revised manuscript.

10. L373: "… controls of the variability *of* C-Q …" I think another "of" needs to be added here.

Revised as suggested. Please refer to line 423 in the revised manuscript.

11. L414: The word "prompt" does not make sense here. I'm not sure what you're trying to convey with that word, but "prompt" doesn't work. Do you mean "prone"?

The comment is right. Corrected. Please refer to line 473 in the revised manuscript.

12. L432: Do you mean "increase" instead of "increment"?

Corrected.Please refer to line 490 in the revised manuscript.

13. L457-458: I generally agree, but it's also important to consider that the Δres50 metric uses INDIVIDUAL grab sample deviations from the long-term C-Q pattern, whereas event-scale metrics like runoff coefficient integrate hydrologic conditions across an entire event. So

accounting for potential biases due to the timing of sample collection and hysteresis become important to consider.

We thank the reviewer for pointing this out. The analysis suggested by the reviewer showed that the relative timing of the sample during events does not affect the long-term average deviations for different event types. Please also refer to our response to Comment 1.

**Reply to Reviewer #2**

14. The paper aims to provide explanation for deviations from long-term C-Q behaviour for different types of hydrological conditions. The authors claim that they are first in doing so, but the only novel thing in this study is a large number of catchments that are investigated. The discussion and implications are pretty much the same as in other studies by the research team, highlighting the incremental character of this study. Thus, to grant the publication of this paper, the authors need to convince the readers about novelty of their work, in light of recent publications in this field.

We apologize that the novelty of our study was not clearly highlighted in the manuscript. The main novelty of our study is in combining the hydrological event classification framework with long-term low-frequency data. To the best of our knowledge runoff event characteristics to explore nitrate dynamics were previously mostly considered in high-frequency studies across individual or few catchments (e.g. Bauwe *et al.*, 2015; Knapp *et al.*, 2020; Heathwaite and Bieroza, 2021), while only Minaudo et al. (2019) and Pohle et al., (2021) considered hydrological conditions using a large sample of catchments and low-frequency data. Combining the information about the hydrological events at the time of sampling with low-frequency data enabled to find systematic deviations in the long-term C-Q relationships induced by different hydrological conditions in a large sample of catchments. Moreover, our large dataset of catchments with contrasting characteristics allows for deducing mechanisms behind the spatial variability of nitrate C-Q deviations across German catchments. Finally, the abundance of low-frequency data worldwide and the transferable nature of the applied event classification framework paves the way to further applications in contrasting environments to better understand scatter in long-term C-Q relationships.

To clarify and highlight the novelty of the study we added/modified the following lines in the revised manuscript.

Low-frequency data combined with runoff event classification in a large number of contrasting catchments

Abstract L17 revised manuscript: *"This study combines a hydrological runoff event classification framework with low-frequency nitrate samples in 184 catchments to explore the role of different runoff events in shaping long-term C-Q relationships and their variability across contrasting catchments. "*

Introduction L63 revised manuscript: "*The scatter of C-Q relationships might also be related to hydrologic conditions at the time of sampling (Knapp et al., 2020, Musolff et al., 2021), which are investigated for a large number of catchments only by a few recent studies (Minaudo et al., 2019; Pohle et al., 2021)."*

Spatial variability of C-Q deviations across contrasting catchments

Abstract L24 revised manuscript*: "Using long-term, low-frequency nitrate data we demonstrate for the first time for a large set of catchments that runoff event types shape observed scatter in long-term C-Q relationships according to the level of hydrologic connectivity characteristic of each runoff event type. In addition, we hypothesize that the level of biogeochemical attenuation taking place in catchments can partially explain the spatial variability of the scatter during different event types."*

Conclusions L520 revised manuscript: "*Moreover, we inferred using catchment descriptors physical mechanisms that possibly explain the spatial variability of this scatter.*"

Systematic deviations of C-Q relationships linked to the runoff event types

Introduction L114 revised manuscript: *"Our study aims for the first time to investigate the presence of systematic deviations in long-term C-Q relationships produced by different runoff event types in a large set of catchments."*

Results L320 revised manuscript: "*We found systematic differences in the direction and magnitude of deviations of nitrate concentrations (Δres50) from the long-term C-Q relationships during different types of runoff events despite the large variety of study catchments (Fig. 5).*"

Transferability of our methods

Discussion L499 revised manuscript: *The abundance of low-frequency data worldwide and the transferable nature of the applied event classification framework provide the means for further applications in contrasting environments to better understand the origins of scatter in long-term nitrate C-Q relationships.*

> 15. I understand that the authors want to show off the contributions from their own team, but there are plenty other papers, not published by your group, that you could refer to in your discussion.

We apologize if we have overlooked relevant references in our manuscript. To show contributions from a larger number of research groups, modified the cited papers by adding or replacing references. We applied the following changes:

L33 revised manuscript: Moreover, due to long-lasting legacy effects a delay in reducing riverine nitrate concentration was reported in many catchments *(Tesoriero et al., 2013; Meter and Basu, 2017; Bieroza et al., 2018; Chang et al., 2021).*

L38 revised manuscript: The shape of C-Q relationships encodes export patterns and reflects the temporally varying quantities of critical substances such as nutrients delivered to streams *(Godsey et al., 2009; Meybeck and Moatar, 2012; Rose et al., 2018).*

L43 revised manuscript: Differences in long-term C-Q-relationships among catchments can be associated with differences in the availability and spatial distribution of solute sources *(Musolff et al., 2017; Dupas et al., 2019; Zhi et al., 2019; Casquin et al., 2021), t*heir hydrologic connectivity *(Seibert et al., 2009; Dupas et al., 2016; Covino, 2017)* and biogeochemical processes within the soil and stream that can retain or permanently remove nitrate from streamwater *(Mulholland et al., 2008; Dupas et al., 2016; Moatar et al., 2017; Benettin et al., 2020).*

L92 revised manuscript: At seasonal scale nutrient transport to streams can be increased with higher hydrologic connectivity in catchments with abundant sources *(Martin et al., 2004; Veith et al., 2020; Guillemot et al., 2021).*

L333 revised manuscript: We argue that during snow-impacted events hydrologic connectivity of sources is high due to elevated wetness conditions (Stieglitz et al., 2003) which is consistent with previously reported high nitrate concentration during the winter period *(Martin et al., 2004; Ocampo et al., 2006; Yang et al., 2018).*

L348 revised manuscript: Rain.dry.uniform and Rain.dry.patchy events occur more often during the dry season when nitrate concentrations are reported to be lower in several studies *(House et al., 2001; Guillemot et al., 2021).*

L433 revised manuscript: Deep sedimentary aquifers have a high potential for denitrification due to a great availability of electron donors, longer transit times, and more anoxic conditions due to sufficient reduction capacity (Kunkel et al., 2004; Wendland et al., 2008; Knoll et al., 2020) producing a lower nitrate supply in deeper soils compared to shallow soil *(Dupas et al., 2016).*

L460 revised manuscript: *Many studies have highlighted the importance of agricultural sources for nitrate*

*export patterns in several catchments (e.g., Moatar et al., 2017; Minaudo et al., 2019; Casquin et al., 2020; Weber et al., 2020).*

L482 revised manuscript: *A reduction in the frequency of snow-impacted events was already shown in Germany over the last decades (Fontrodona Bach et al., 2018; Chan et al., 2020; Taszarek et al., 2020).*

Specific comments

16. Line 16 grammar

Thank you for this suggestion. We will modified the text.

L14 revised manuscript: *"Although previous studies investigated the origins of this scatter in individual or in a few catchments, the role of different runoff event types across a large set of catchments is not yet fully understood."*

17. Line 16 how about Winter et al? This topic seems to have been already covered by your colleagues, so what is the novel aspect of this study? There have been also other paper studying how different storm event response contribute to scatter in C-Q data making this statement untrue, please update the list of previous studies on the topic in the introduction

There are in fact considerable differences between Winter et al. (2021) and this manuscript. The work of Winter et al. focuses on the variability between runoff events during a limited 4-years period considering only samples takan during runoff events and using high-frequency data in only 6 neighboring catchments in Central Germany. The study finds that variability of hysteresis patterns decreases from runoff events induced by rainfall with dry antecedent conditions to snow-impacted events. In contrast, this manuscript uses low-frequency across 184 catchments data and investigates the effect of runoff event types in long-term C-Q relationships. The increase of nitrate concentration during snow-impacted and the decrease decrease during rainfall events with dry antecedent conditions of Winter et al. (2021) is also confirmed in our work. However, a much larger number of catchments with contrasting characteristics used in this study allow us to investigate systematic nitrate deviations from long-term C-Q relationships across catchments and attribute spatial patterns of deviations to potential physical mechanisms using catchment characteristics. We modify L14 of the revised manuscript to clarify the differences between the two studies.

L14 revised manuscript: *"Although previous studies investigated the origins of this scatter in individual or in a few catchments, the role of different runoff event types across a large set of catchments is not yet fully understood."*

We apologize if we have overlooked relevant papers on the scatter in C-Q relationships due to the different storm responses. We have added now additional references to the Introduction.

L67 revised manuscript: The cause of this scatter can also be traced to a variety of responses observed at the event-scale in several studies with high-frequency data in single or a few catchments *(e.g., Bowes et al., 2015; Lloyd et al., 2016; Koenig et al., 2017; Gorski and Zimmer, 2021).*

L74 revised manuscript: Disparate patterns of the event C-Q relationships in a catchment over time are mainly attributed to varying dominant flow sources (e.g., groundwater, shallow subsurface flow), antecedent wetness conditions (Inamdar et al., 2006; Vaughan et al., 2017; Knapp et al., 2020), *time of fertilizer application (Bowes et al., 2015; Dupas et al., 2016; Outram et al., 2016), biogeochemical cycling (Heathwaite and Bieroza, 2021)* and runoff event characteristics or types *(Butturini et al., 2006; Bauwe et al., 2015; Chen et al., 2020; Knapp et al., 2020; Heathwaite and Bieroza, 2021).*

18. Line 22 'indicating low nitrate concentrations' – this does not make sense

Thank you for pointing this out. We will modified the text to clarify this issue:

L20 modified manuscript: *"In contrast, negative deviations occur mostly for rainfall-induced events with dry antecedent conditions, indicating the occurrence of lower nitrate concentrations in river flows than their long-term pattern values during this type of events."*

> 19. It is not clear if you analyse high-frequency or low-frequency C-Q data, this should be clarified at the very beginning of the paper. Without this information it is difficult to judge the quality of your hypotheses.

Thank you for the comment. We agree that this should be clarified in the Introduction section. We added the following sentence in line 71 in the revised manuscript:

*"Our study relies on low-frequency nitrate data, which is often used to build long-term C-Q relationships (e.g. Cartwright et al., 2020, Diamond and Cohen 2018). However, studies with high-frequency data found large variability in the C-Q patterns during events (e.g. Knapp et al., 2020; Dupas et al., 2016; Vaughan et al., 2017) that might add scatter to the long-term C-Q relationship.*

> 20. Figure 1 should be part of methods or results but not introduction

Thanks for the suggestion. Since we consider this Figure essential to set up our hypothesis and do not use it in our Method section, we would prefer to keep it in the Introduction section.

> 21. Hypothesis 1 is not clear. Do you mean individual C-Q points?

Thank you for pointing this out. To clarify this we will modify it as follows:

L121 revised manuscript: *"1 Do samples collected during different event types deviate differently from the long-term C-Q relationships observed at the catchment outlets?"*

> 22. Not clear how daily discharge data can provide information about short storm events with duration of hours?

Thank you for your question. We use daily streamflow to identify events. This implies that the shortest event that can be captured has a duration of at least 1 day. Any event shorter than 1 day cannot be captured with the available data. We will modified the following sentence in Line 156 of the revised manuscript:

*"The method includes baseflow separation, precipitation attribution and an iterative procedure to adjust site-specific thresholds for the refinement of multi-peak events. We use daily streamflow data to identify events. This implies that only events longer than 1 day are captured."*

> 23. In this sense, using a term 'event classification' is misleading. I would rather use classification of 'hydrological conditions'.

We prefer to keep the term "event classification" instead of "hydrological conditions" as the former more accurately represents the information combined in the event types and is a standard in the hydrological literature (e.g., Bauwe et al., 2015; Chen et al., 2020; Ross et al., 2019; Xie et al., 2019). Apart from information on hydrological conditions often used (i.e., wetness conditions) it also includes information on the nature of precipitation events and spatial distribution of soil moisture.

> 24. Since you have low-frequency samples they are sampled randomly over the hydrograph. So samples that belong to the same hydrological condition can have been sampled on a rising, falling limb of the hydrograph or baseflow conditions. Thus, some of your scatter in each hydrological condition group can be attributed to when on the hydrograph your samples were taken. Please clarify.
>
> I have just noticed that Reviewer 1 expressed similar concerns regarding the role of C-Q

hysteresis. This is a key weakness of your approach.

We appreciate the reviewer's comment on the possible effect of hysteresis at the event scale. We agree that this effect requires additional attention in the manuscript. The first reviewer have raised a similar concern, please refer to Comment 1.

References:

[revised manuscript text omitted]

---

## Author Response (AR2)

Manuscript Egusphere-2022-205

Disentangling Scatter in Long-Term Concentration-Discharge
Relationships: the Role of Event Types

We acknowledge the Editor and two Reviewers for providing detailed and constructive comments to our manuscript in the present and previous rounds of review. This document provides our point-by-point replies (black color) to the reviewer comments (blue color). In addition, we enclosed a revised manuscript with track changes in our response.

**Reply to Editor comments**

Thank you for thoroughly addressing the comments raised by reviewers in the last round of review. In the present round, one reviewer raised additional comments, so I am returning your manuscript for minor revision. I look forward to receiving your revised manuscript.

Thank you very much for managing the review process. In the following lines, we address all the new comments of the reviewers.

**Reply to Reviewer #2**

General comments

Saavedra et al. provide a comprehensive study on nitrate concentration-discharge relationships. With an original methodology, they successfully identify key relationships between C-Q patterns and hydrological event types. The approach is sound, the results and interpretations are well conducted and interesting for the readership of HESS. I have raised some relatively minor issues, which must be properly addressed prior to publication.

We appreciate the reviewer's constructive comments and suggestions. We address all your comments in the following lines.

Major comments:

L139: could you please represent these 4 regions on the map (Fig 2a)? For someone outside Germany, this could be useful to better understand the interpretation from Fig 5a.

Thanks for the insightful comment. We added in Fig. 2 the four natural regions of Germany.

L177: "neutral (b~0) indicates a weak dependency of C and Q.". This could also be because the long-term CQ relationship is the combination of different opposing events or seasonal patterns vs storm events: for instance, at the seasonal scale there is a dilution pattern, which combines with enrichment during storm events. Since the entire study here has for objective to disentangle the scatter in C-Q plots, we expect from the authors to carefully chose their words in this particular case. Please, consider rephrasing.

We agree with the comment and we rephrased the sentence. Please refer to line 176 in the revised manuscript.

L187-197: this methodological approach is absolutely key for the robustness of the entire study, because a different choice (for instance compute deviation from CQ relationship from non-event observations) would likely lead to different outcomes. It is unclear to me where to find the results from this important bootstrapping analysis. Please make the result from this analysis much clearer as the reader shouldn't have any doubt about your methodological choices.

Thanks for the insightful comment. We added bootstrapping results to supplementary material. Please refer to Figure S2 and lines 193 in the revised manuscript.

L312: "Event runoff coefficients exhibit a larger variability across event types than across catchments for most of the catchments.". Please place this sentence after the following one "Catchment median event runoff coefficients exhibit a coefficient of variation of 41% across catchments. Nevertheless, median runoff coefficients of event types exhibit coefficients of variation in different catchments from 12% to 118%, with a median value of 67% across catchment" as it is very hard to get as it is now.

We agree with the comment and we rephrased the sentence. Please refer to lines 309-313 in the revised manuscript.

L344-346: "Lower nitrate concentration during runoff events with dry antecedent conditions can be explained by low pre-event conditions linked to hydrological and biogeochemical drivers in addition to possible dilution during runoff events.". What are exactly these "hydrological and biogeochemical drivers" being mentioned? Please be specific and name directly these processes.

Thanks for the insightful comment. We specified each driver and rephrased the whole paragraph. Please refer to lines 337-355 in the revised manuscript.

L353: "due to a more efficient removal". Please explain the driver behind, because a more efficient removal necessarily removes more nitrate! Please be more specific with the name of the processes behind (e.g. denitrification, biological uptake, … etc).

Thanks for the insightful comment. We rephrased the whole paragraph to be more specific. Please refer to lines 337-355 in the revised manuscript.

L378-379: "Most of nitrate samples during no event conditions coincide with low rates of discharge (Fig. 4a) as well as Rainfall events with dry antecedent conditions (i.e., Rain.dry.patchy and Rain.dry.uniform).". How could a sample taken under no-event condition coincide with some rainfall events? Please clarify or revise.

We thank the reviewer for pointing this out. We explain in this sentence that samples taken during Rainfall events with dry antecedent conditions and samples taken during no-event conditions are taken at relatively low discharge levels. We rephrased the paragraph for clarification. Please refer to lines 365-369 in the revised manuscript.

L439-440: "We acknowledge that catchment characteristics might be highly correlated (Fig. S4).". It is indeed needed to do so, but this sentence seems lost in the paragraph, although one can understand with the end of the paragraph what was meant. Please revise this sentence, because the reader should not have to read the end of the paragraph to understand the meaning of the sentences found mid-paragraph.

Thank you for the comment. We modified and reorganized the paragraphs to improve readability. Please refer to lines 436-450 in the revised manuscript.

Minor comments:

L29: I'm not native English speaker, but are you sure "portend" is appropriate in this sentence?

We changed the sentence. Please refer to line 29 in the revised manuscript.

L30: "particularly of nitrate". I agree excessive nitrate concentrations are partly responsible, but please, temper this statement as P is in general the main driver for eutrophication in freshwater-ecosystems.

We modified the sentence. Please refer to line 30 in the revised manuscript.

L55: "The time of fertilizer". Consider "timing" instead of "time"

Revised as suggested. Please refer to line 55 in the revised manuscript.

L57: "On the other hand," Please remove

Removed. Please refer to line 57 in the revised manuscript.

L83: "C-Q relationships are more positive due to the accumulation in soil during dry periods of nitrate from atmospheric deposition". Consider rephrasing to "C-Q relationships are more positive due to the accumulation of nitrate in the soil during dry periods by atmospheric deposition"

Revised as suggested. Please refer to line 82 in the revised manuscript.

L84: "Eurpe," Typo. Please correct

Corrected. Please refer to line 83 in the revised manuscript.

L167: "This implies that only events longer than 1 day are captured.". Can we reliably detect a hydrological event of 2 days with daily data? Is there a minimum number of daily observations needed to detect an "event"?

We thank the reviewer for pointing this out. The event identification procedure consider events only if there is an inducing snowmelt or rainfall event occuring before the event (using seasonal basin lag times as searching window prior the runoff event) and if the increase in discharge is at least 10% compared to the baseflow discharge rate. The minimum event duration is therefore 1 day. For further information, please refer to Tarasova et al., 2018. Moreover, in our dataset the 95% of the identified events exhibit a duration of 3 or more days. We included this information in the revised manuscript, please refer to lines 155-156.

L248: Please insert a "the" in between "considerable scatter in" and "regressions"

Corrected. Please refer to line 247 in the revised manuscript.

L276: Please guide the reader, this paragraph is a bit lost in between two bigger paragraphs and the reader needs to read the end of it to understand what it is about (i.e. the timing of sampling during rising/falling limb of the hydrograph).

Thank you for the comment. We modified and reorganized the paragraph to improve readability. Please refer to lines 272-277 in the revised manuscript.

L277: "Fig. S6b" Please sort the figures numbering from SI accordingly to their appearance in the text.

Thanks for the insightful comment. We sorted the figures numbering from Supplementary Material in the revised manuscript.

L278-281: this sentence is very long and difficult to read. Please simplify to help the reader.

Thank you for the comment. We rephrased the paragraph. Please refer to lines 272-277 in the revised manuscript.

L285-286: maybe give numbers to make it clearer?

Thank you for the comment. We added the values of interquartile ranges in the revised manuscript. Please revise lines 279-282 in the revised manuscript.

L295: "however the last feature shows" which feature is being mentioned? Please be more specific.

Thank you for the comment, corrected as suggested. Please refer to line 291 in the revised manuscript.

L308: "A nitrate surplus is strongly related only to Rain.wet residuals." Please revise to "Nitrate surplus is significantly related only to Rain.wet residuals."

Corrected. Please refer to line 305 in the revised manuscript.

L338: "movilized". Please revise

Corrected. Please refer to line 334 in the revised manuscript.

L339: "In adition, the movilized water during this events is less affected by biogeochemical processes due to lower microbial activity induced by low temperature during snow-impacted events.". Could you add at least 1 reference to support this statement?

We added a reference. Please refer to line 335 in the revised manuscript.

L341-374: could you make this a single paragraph? Splitting the discussion in so many short paragraphs makes it difficult to follow.

We merged the paragraphs. Please refer to lines 337-363.

L355-356: "Moreover, during runoff events with dry antecedent conditions nitrate concentrations can decrease below pre-event concentration level.". Isn't it redundant with previous statements in previous paragraph?

We agree with the comment we rephrased the paragraph. Please refer to lines 337-363.

L449-454: these sentences are a repetition of the lines 439-444. Please delete and make sure this paragraph still makes sense.

We agree with the comment. We modified the paragraph. Please refer to lines 436-450.

L481: "a substantial decline in seasonal snowpack accumulation and earlier snowmelt onset in Central Europe". Yes but wouldn't the frequency of rain on snow events increase? Please make sure of this aspect, which would completely change the interpretation of the outcome paragraph.

Thanks for pointing this out. To the best of our knowledge, there is a consensus in future and present negative trends in snow accumulation in the study region. This negative trend is consistent with a decreasing trend observed in the frequency of rain-on-snow events (Cohen et al., 2015). We added references and modified the text. Please refer to lines 461-466 in the revised manuscript.

L519-520: "Moreover, we inferred using catchment descriptors physical mechanisms that explain the spatial variability of this scatter.". Please reorder or revise, something is missing or misplaced in this sentence.

We agree with the reviewer. We deleted the sentence for readability.

References:

Cohen, J., Ye, H., & Jones, J. (2015). Trends and variability in rain-on-snow events. *Geophysical Research Letters*, *42*(17), 7115–7122. https://doi.org/10.1002/2015GL065320

Tarasova, L., Basso, S., Zink, M., & Merz, R. (2018). Exploring Controls on Rainfall-Runoff Events: 1. Time Series-Based Event Separation and Temporal Dynamics of Event Runoff Response in Germany. *Water Resources Research*, *54*(10), 7711–7732. https://doi.org/10.1029/2018WR022587